# Elastocapillary sequential fluid capture in hummingbird-inspired grooved sheets

Emmanuel Siéfert [1,2] ✉, Benoit Scheid[3], Fabian Brau [1] & Jean Cappello [3,4] ✉

Passive and effective fluid capture and transport at small scale is crucial for industrial and medical applications, especially for the realisation of point-of-care tests. Performing these tests involves several steps, including capturing biological fluid, aliquoting, reacting with reagents, and reading the results. Ideally, these tests must be fast and offer a large surface-to-volume ratio to achieve rapid and precise diagnostics with a reduced amount of fluid. Such constraints are often contradictory as a high surface-to-volume ratio implies a high hydraulic resistance and hence a decrease in the flow rate. Inspired by the feeding mechanism of hummingbirds, we propose a frugal fluid capture device that takes advantage of elastocapillary deformations to enable concomitant fast liquid transport, aliquoting, and high confinement in the deformed state. The hierarchical design of the device – that consists in vertical grooves stacked on an elastic sheet – enables a two-step sequential fluid capture. Each unit groove mimics the hummingbird's tongue and closes due to capillary forces when a wetting liquid penetrates, yielding the closure of the whole device in a tubular shape, in the core of which additional liquid is captured. Combining elasticity, capillarity, and viscous flow, we rationalise the fluid-structure interaction at play both when liquid is scarce and abundant. By functionalising the surface of the grooves, such a passive device can concomitantly achieve all the steps of point-of-care tests, opening the way for the design of optimal devices for fluid capture and transport in microfluidics.

When a porous medium is put in contact with a wetting liquid, capillary forces lead to the imbibition of the liquid inside the pores of the material, overcoming gravity[1]. This phenomenon of capillary rise is ubiquitous: it brings water to the upper layer of soils[2], drives sap in plants[3], and plays a crucial role in the textile and paper industries[4,5]. Fundamental capillarity laws were obtained a century ago by Lucas[6] and Washburn[7], by balancing the driving capillary force and viscous friction in horizontal pores. This basic modelling approach has been further enriched in a vast literature, to model the capillary rise in pipes of various orientations with gravitational effects[8,9] and various closed[10,11] or open[12–15] geometries, to regularise the early dynamics via inertial[16,17] or contact line friction[18] effects, and to model the effects of

non-Newtonian liquids[19,20]. Recently, bioinspired asymmetric textures were designed to obtain directional capillary transport along specific directions[21,22]. However, the imbibing speed is set by the typical pore size, yielding slow liquid penetration at small scales, that are particularly relevant for medical testing devices since they offer a large surface-to-volume ratio[23]. To resolve these conflicting physical constraints and obtain fast penetration in confined pores, one possible approach relies on passively changing the pore shape and size using a soft material, such that the pores are relatively larger in their undeformed state, promoting fast liquid imbibition, and smaller in their deformed state, offering a confined environment beneficial for testing purposes. Soft porous solids, that may be deformed by capillary forces

[1]Nonlinear Physical Chemistry Unit, Université Libre de Bruxelles (ULB), Bruxelles, Belgium. [2]LIPhy, CNRS, Université Grenoble Alpes, Grenoble, France. [3]Transfers Interfaces and Processes, Université Libre de Bruxelles (ULB), Bruxelles, Belgium. [4]Institut Lumière Matière, CNRS, Université Claude Bernard Lyon 1, Villeurbanne, France. ✉e-mail: emmanuel.siefert@univ-grenoble-alpes.fr; jean.cappello@ulb.be

leading to a coupling between the deformable porous medium and the liquid[24–26], are ideal candidates to meet these needs. Yet, so far, studies mostly focused on the coalescence of the pores, leading to an enhanced but slower capillary rise[27,28].

Hummingbird tongues offer an inspiring natural example of a deformable structure that meets the needs of fast liquid capture. To feed, hummingbirds need to hover above flowers[29], the most energy-consuming form of locomotion in the animal kingdom[30]. The feeding strategy of this small nectarivore thus has to be fast and efficient. Evolution led to the unique shape of the hummingbird tongue: it is composed of two flexible curved lamellae fixed to a supporting rod and forming an open groove (Fig. 1a). In vivo observations reveal that these grooves act as dynamic elastocapillary liquid-trapping devices[31] that passively close into tubes as it is withdrawn from an abundant nectar source. When nectar is scarce however, the tongue efficiently acts as a capillary syphon[32–34], leading to a fast capillary rise in the closing grooves that may be actively enhanced by a micropump effect[35].

Inspired by hummingbirds, we go one step further and design flat architected elastic sheets (right panels of Fig. 1a) that contain many vertical grooves, each of them acting as the curved lamella of the hummingbird's tongue. This metatongue takes advantage of the collective elastocapillary deformation of each groove to deform at the device scale. Details on the fabrication of these structures are given in 'Methods' section 'Elastomer preparation'. Depending on the amount of liquid available, the device is either slightly put in contact or fully dipped into the liquid ('Methods' section 'Experimental apparatus' and Supplementary Fig. 1).

When liquid is scarce, as contact is made with the liquid, a first capillary rise occurs in the grooves (first instants in Fig. 1b). This first capillary rise of height $\ell_I$ induces a capillary torque on the bottom sheet of each groove and hence its bending. This bending brings adjacent walls closer and closes each groove into a pipe of circular sector cross-section, trapping the liquid as in the case of the hummingbird tongue. As each groove deforms, the entire sheet consequently bends and may eventually close into a tubular shape, leading to a second capillary rise $\ell_{II}$ in the core of the newly formed tube (final stages in Fig. 1b, Supplementary Videos 1 and 2). The dynamics and final height of this second capillary rise are intrinsically related to the groove geometry as the radius of the newly formed tube is given by the

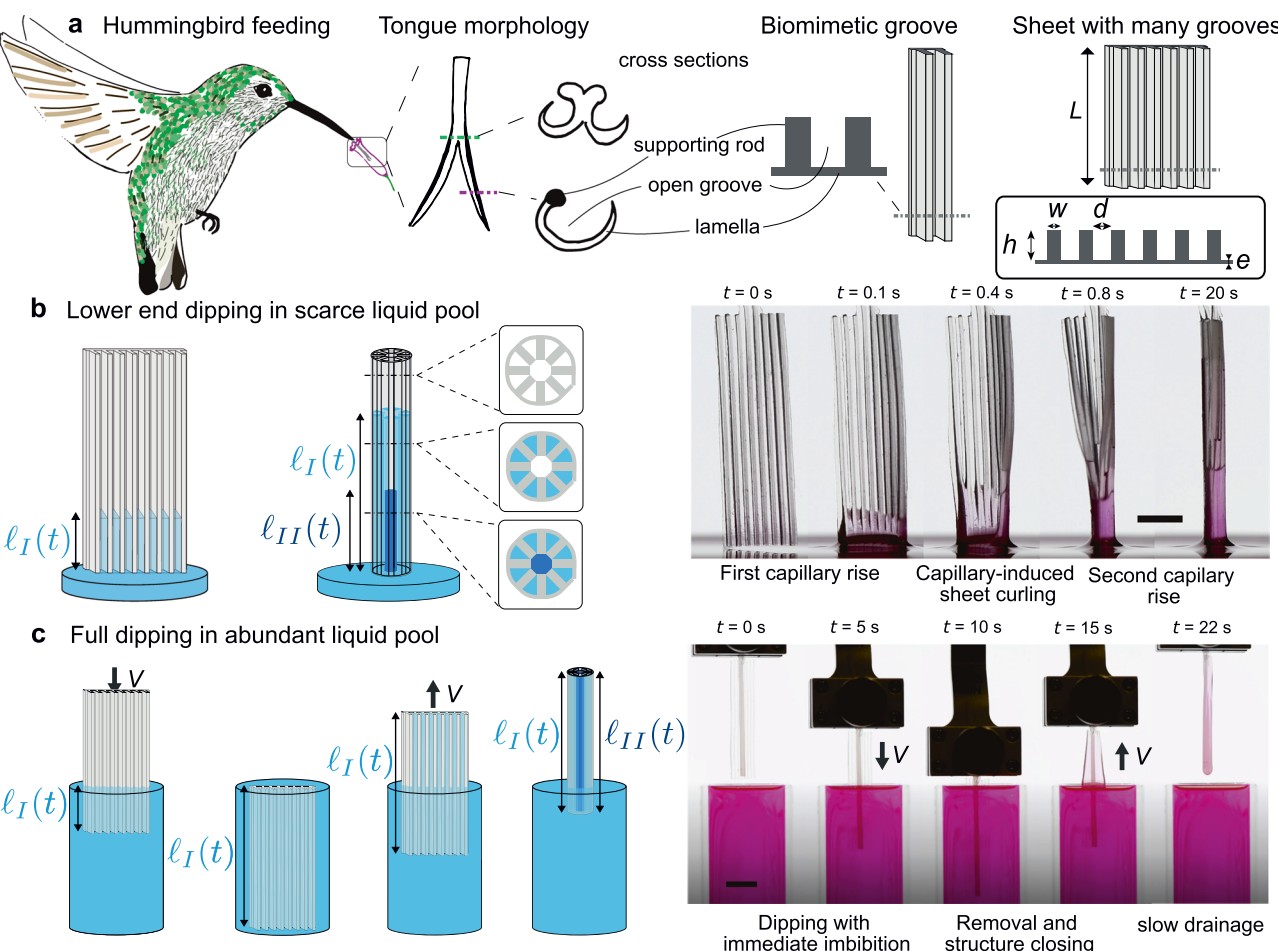

**Fig. 1 | Hummingbird-inspired fluid capture. a** Hummingbirds feed using their flexible, grooved tongues. Inspired by this feeding mechanism, we created a biomimetic device with multiple stacked grooves, forming a groovy sheet of length $L$, thickness $e$, decorated with walls of height $h$, width $w$, forming grooves of width $d$. **b** When the lower end of the device is dipped into a wetting fluid bath, a first capillary rise (characterised by the front elevation $\ell_I(t)$) occurs inside the grooves, inducing a capillary torque on the sheet that bends into a tubular shape. A second capillary rise ($\ell_{II}(t)$) takes then place in the core of the newly formed tube. Snapshots of the hierarchical capillary rise in a grooved sheet ($e = 135 \pm 10\,\mu m$, $h = 800 \pm 50\,\mu m$, $w = 400 \pm 40\,\mu m$, $d = 700 \pm 40\,\mu m$, $L = 26.1 \pm 0.2\,mm$) made of a silicone rubber. Liquid used is silicone oil V10 ($\mu = 9.5 \times 10^{-3}$ Pa.s, $\gamma = 0.021$ N/m, $\rho = 930$ kg.m$^{-3}$). Scale bar: 5 mm (see Supplementary Videos 1 and 2). **c** When the liquid is abundant, the structure is fully immersed in the liquid pool. As the structure is dipped, at a velocity $V$, liquid enters the grooves. When removing the device, the structure deforms and closes in a tubular shape in which additional liquid is captured. The snapshots show a flexible grooved sheet ($L = 40 \pm 0.2$ mm, $h = 800 \pm 20\,\mu m$, $w = 360 \pm 20\,\mu m$, $d = 700 \pm 50\,\mu m$, $e = 135 \pm 10\,\mu m$) as it is dipped and pulled out from a bath (silicone oil V1000, $\mu = 0.96$ Pa.s, $\rho = 960$ kg.m$^{-3}$, $\gamma = 0.021$ N/m) at constant speed $V = 200$ mm/min. Scale bar: 1 cm.

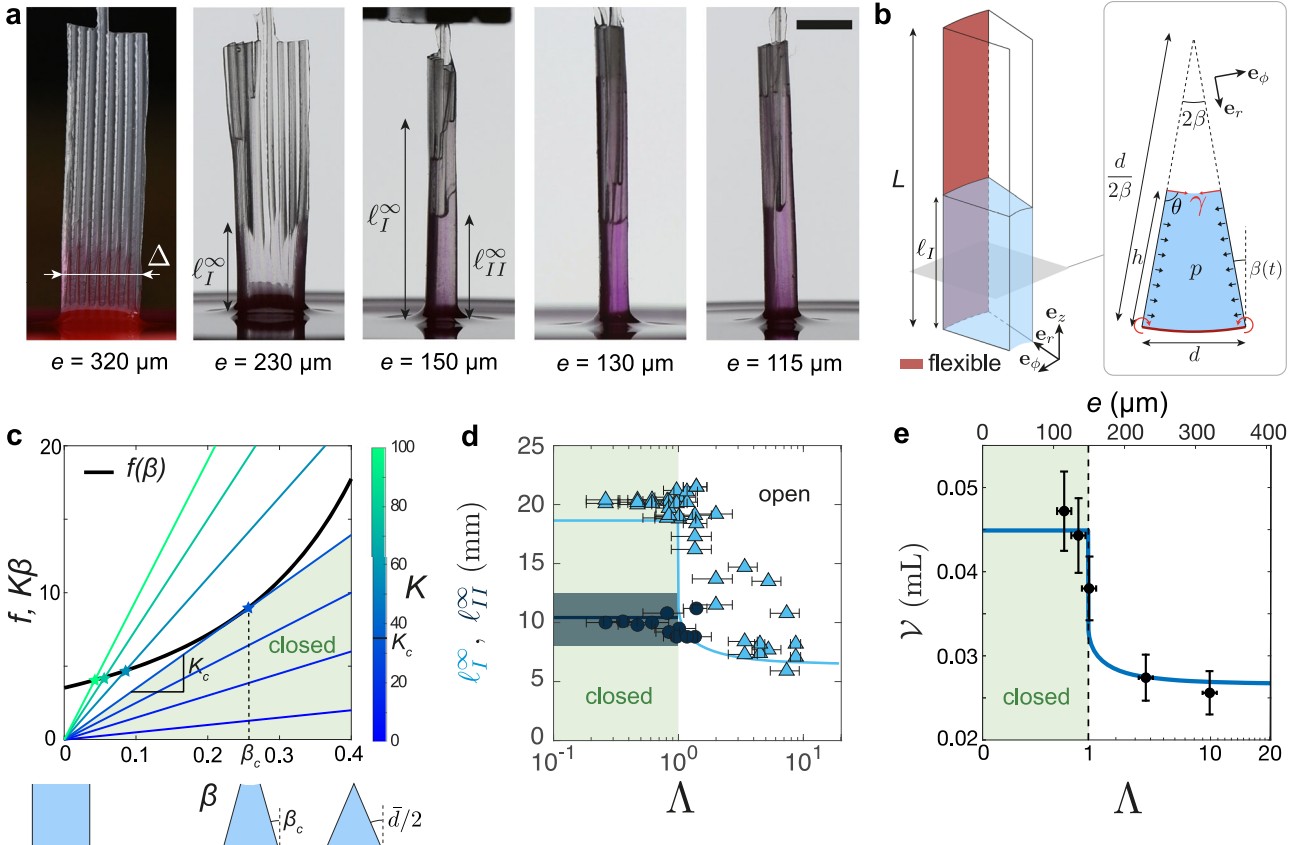

**Fig. 2 | Equilibrium state. a** Snapshot of the final equilibrium shape of the ribbed structure for various sheet thicknesses $e$ with the same groove geometry: $h = 810 \pm 40\,\mu m$, $w = 400 \pm 40\,\mu m$, $d = 700 \pm 50\,\mu m$. A transition to closure is observed for sufficiently thin sheets. Scale bar: 5 mm. **b** Schematics of the flexible groove model: the capillary rise $\ell_I$ in the groove induces a capillary torque acting on the bottom flexible sheet of length $L$ (in red) via the walls considered as rigid, inducing a radius of curvature $R = d/(2\beta)$. **c** The equilibrium closing angle corresponds to the intersection between the curves $f(\beta)$ and $K\beta$ (defined in Eq. 4). Below a critical value $K = K_c$ no solutions are possible and the structure close completely forming a tube (green area). **d** Final equilibrium elevations $\ell_I^\infty$ and $\ell_{II}^\infty$ as a function of the dimensionless transition parameter $\Lambda$ defined by Eq. (5) for devices with the same groove geometry (given in **a**) but with varying length $L$ and thickness $e$. For both capillary rises, a sharp transition occurs at $\Lambda \sim 1$. Uncertainties are estimated using the theory of error propagation. **e** Total captured volume $\mathcal{V}$ as a function of the dimensionless transition parameter $\Lambda$ and of the sheet thicknesses $e$. Experiments corresponding to the five structures shown in (**a**) (black dots) are compared to the theoretical prediction (blue line). Structure closing is associated to a fast increase of the captured volume. Uncertainties are estimated using the theory of error propagation. Source data are provided as a Source Data file.

curling of the entire structure that stops when self-contact between the neighbouring walls occurs[36].

When liquid is abundant, a second liquid capture strategy consists in fully dipping the grooved sheet in the bath before withdrawing it (Fig. 1c). As the structure is immersed, liquid enters the grooves from the side, and the structure gets immediately imbibed. When removing the structure from the bath, capillary forces lead to its closing into a tube and hence a liquid capture in the grooves and in the newly formed tube. During the withdrawing phase, the liquid drains from the pipes. Hence, the amount of captured liquid will depend on the ratio between the withdrawing velocity and the draining speed.

## Results

### Condition for groove closure

We start by considering the situation where the liquid is scarce, and the grooved sheet is slightly dipped in the liquid bath. Depending on the geometry of the grooved sheet (groove height $h$ and width $d$, wall width $w$ and sheet thickness $e$; see Fig. 1a) and on its mechanical properties (silicone rubber of Young's modulus $E$ and Poisson ratio $\nu$) three main scenarios can be observed while capillary rise occurs: (i) the structure remains undeformed and the capillary rise takes place in undeformed grooves of rectangular cross-section (first panel of

Fig. 2a), (ii) the sheet slightly bends due to capillary forces and the capillary rise occurs in an open channel (second panel of Fig. 2a), (iii) the sheet bends until neighbouring walls are in contact leading to the closure of the structure into a tube (last three panels in Fig. 2a). We aim at rationalising these three cases depending on the parameters of the system.

In the limit of a thick sheet (large $e$) or large Young modulus, i.e. when the sheet deformation is negligible (case (i)), the system is equivalent to the capillary rise inside rigid open grooves of depth $h$ and width $d$, that has been at the core of recent studies[13–15]. In this configuration, the liquid rises up to an equilibrium height $\ell_I^{\infty,\,open}$ that results from the competition between the gravitational force $\mathbf{F}_g = -\rho g h d \ell_I^{\infty,\,open} \mathbf{e_z}$, with $\rho$ the fluid density and $g$ the gravitational acceleration, and the pulling capillary force, that is shown to barely depend on the depth $h$ of the channel as long as there is total wetting (i.e. $\theta_Y = 0$ with $\theta_Y$ being the contact angle) and that reads $\mathbf{F}_\gamma = h d \Delta p \mathbf{e_z}$, where $\Delta p = 2\gamma/d$ is the pressure drop due to the curvature of the liquid interface and $\gamma$ is the surface tension. Under these conditions, the equilibrium height is

$$\ell_I^{\infty,\,open} = \frac{2\ell_c^2}{d}, \qquad (1)$$

with $\ell_c = \sqrt{\gamma/(\rho g)}$ being the capillary length. Note that this elevation is equivalent to the equilibrium capillary rise between two infinite parallel plates separated by a distance $d$.

As mentioned above, the capillary rise in the grooves induces a torque on the sheet through the walls, and, when the device is flexible enough, a bending of a sheet occurs leading to partial (cases (ii)) or complete (cases (iii)) closing of the groove. We model this closure by a single parameter, a closing angle $\beta$ (Fig. 2b). We thus assume that the sheet bends along an arc of circle and consider that the deformation of the sheet is identical along the structure length. This assumption is valid as long as the curvature persistence length−defined as the length from which a curvature imposed at an extremity completely disappears−is large compared to the length of the structure itself[37]. We also assume that the vertical air-liquid interface joining the wall free ends is bent along an arc of circle of radius $R-h$ with the same central angle $2\beta$ as the sheet. The angle $\theta$ between the liquid interface and the walls at the wedges is thus assumed to be constant and equal to $\pi/2$ at any vertical elevation (Fig. 2b).

As $\beta$ increases, the cross-section of the groove, $A(\beta, \bar{d}) = h^2 (\bar{d} - \beta)$, with $\bar{d} = d/h$ the groove aspect ratio, and the distance between the top corners of neighbouring walls decrease, resulting in increased confinement and hence increased capillary force, that reads $\mathbf{F}_\gamma = 2\gamma(1+\beta) h \mathbf{e_z}$ (see 'Methods' section 'Pressure difference in a groove' for a detailed derivation). The gravitational force in the liquid column reads: $\mathbf{F}_g = -\rho g A(\beta, \bar{d}) \ell_l \mathbf{e_z}$, where $\ell_l$ the liquid column height in the groove. Balancing these two forces yields the equilibrium capillary rise:

$$\ell_l^\infty = \frac{2\ell_c^2}{h} \frac{(1+\beta)}{\bar{d} - \beta}, \quad \bar{d} = d/h. \qquad (2)$$

To fully determine this length, we need to compute the closing angle $\beta$ that is set by mechanical balance on the sheet. The flexible sheet between neighbouring walls is subjected to a bending moment that has two contributions. A first contribution comes from the depression in the rising liquid column, which generates a total force acting at the middle of the wall: $\mathbf{f}_p = \pm h \int_0^{\ell_l} p(z)dz \mathbf{e_\phi}$, where $p(z) = F_\gamma z/[A\ell_l]$ is the linearly varying pressure within the liquid column and where $+$ (resp. $-$) sign holds for the force acting on the left (resp. right) wall. Considering a lever arm $\mathbf{l}_p = -h/2\mathbf{e_r}$, the force produces a moment: $\mathbf{M}_p = \mathbf{l}_p \wedge \mathbf{f}_p = \mp h \int_0^{\ell_l} p(z)dz \mathbf{e_z} h/2$, where $-$ (resp. $+$) sign holds for the left (resp. right) end of the sheet. The second contribution to the torque comes from the contact line at the top corners of the walls that leads to a force $\mathbf{f}_\gamma = \pm \gamma \int_0^{\ell_l} \sin\theta \, dz \mathbf{e_\phi}$, $+$ (resp. $-$) sign holds for the force acting on the left (resp. right) wall. The lever arm being $\mathbf{l}_\gamma = -h\mathbf{e_r}$, the force leads to the moment $\mathbf{M}_\gamma = \mathbf{l}_\gamma \wedge \mathbf{f}_\gamma = \mp h\gamma \int_0^{\ell_l} \sin\theta \, dz \mathbf{e_z} \simeq \mp \gamma h \ell_l \mathbf{e_z}$, where $-$ (resp. $+$) sign holds for the left (resp. right) end of the sheet. Note that for the sake of simplicity, we assume $\theta = \pi/2$, hence the simplified expression of $\mathbf{M}_\gamma$. Neglecting the sheet inertia, the elastic restoring torque $\mathbf{M_B}$ is given by:

$$\mathbf{M}_B = \frac{B}{R} \mathbf{e_z} = \mathbf{M}_p + \mathbf{M}_\gamma, \qquad (3)$$

where $B = Ee^3L/[12(1-v^2)]$ is the bending modulus of the flexible sheet, $E$ and $v$ the Young modulus and the Poisson ratio of the material, respectively, and $R = d/(2\beta)$ the radius of curvature of the sheet. Note that the forces $\mathbf{f}_\gamma$ and $\mathbf{f}_p$ act on each side of the walls and thus compensate, leading to no deflection of the walls. Substituting the moments expressions in Eq. (3), taking the final capillary rise (Eq. 2) and projecting into the $z$-axis yields (see 'Methods' section 'Torque balance' for a detailed derivation):

$$K\beta = \left[2 + \frac{1+\beta}{(\bar{d} - \beta)}\right] \frac{(1+\beta)}{(\bar{d} - \beta)} \equiv f(\beta, \bar{d}), \qquad (4)$$

where $K = 2B/[\gamma d\ell_c^2]$ is the dimensionless stiffness of the sheet. The left-hand side of Eq. (4) is linear in $\beta$, whereas the right-hand side is more complex and is plotted in Fig. 2c for $\bar{d} = 0.8$. The solution of this equation can be obtained by a graphical construction (see Fig. 2c), looking at the intersection between both curves. When $K$ is large, the flexible sheet is stiff and barely bends ($\beta$ stays close to 0) and, consequently, $\bar{\ell}_l^\infty$ tends to the equilibrium height in an open groove (Eq. 1). As $K$ decreases, the equilibrium value of $\beta$ increases until $K$ reaches a critical value $K_c$ below which there is no solution for Eq. (4) in the physical interval $0 \le \beta^\infty \le \bar{d}/2$, ($\beta = \bar{d}/2$ corresponding to self-contact between neighbouring walls and complete closure of the groove). Hence, for $K < K_c$, the groove closes until additional contact forces appear for $\beta = \bar{d}/2$ to balance Eq. (4) (case (iii)).

The condition $K < K_c$ can be rewritten as the following inequality for the groove closure up to order 2 in $\bar{d}$ (see 'Methods' section 'Stationary solution and condition for groove closure' for a detailed derivation and Supplementary Fig. 3):

$$\Lambda \equiv \frac{8Bd^2}{27\gamma h^3 \ell_c^2} \left[1 + 2\bar{d} + 0.556\bar{d}^2\right]^{-1} < 1. \qquad (5)$$

When this criterion is fulfilled, the equilibrium height of the first capillary rise increases to $\ell_l^{\infty, \text{closed}} = \ell_l^\infty(\beta = \bar{d}/2) = \ell_l^{\infty, \text{open}} (2 + \bar{d})$, which is more than twice the elevation in the rectangular open groove (Eq. 1). By choosing the proper width of the sheet, or equivalently the right number of grooves $n \approx 2\pi h/d$, this situation of self-contact between neighbouring walls corresponds to the closure of the whole structure into a tube of inner radius $R_{int} = hw/d$[36]. The second capillary rise $\ell_{II}$ (Fig. 2a) reaches the classical Jurin height, $\ell_{II}^\infty = 2\ell_c^2/R_{int}$, at equilibrium (see 'Methods' section 'Second capillary rise dynamics').

The validity of the closure criterion (Eq. 5) can be assessed in Fig. 2d where experimental measurements (symbols) of the equilibrium heights of the first (light blue triangles) and second (dark blue circles) capillary rise are shown as a function of the dimensionless transition parameter $\Lambda$. Indeed, experimental measurements show a sharp transition at $\Lambda \simeq 1$ for both capillary rises $\ell_I$ (from $\ell_I^{\infty, \text{open}}$ to $\ell_I^{\infty, \text{closed}}$) and $\ell_{II}$ (from 0 to $\ell_{II}^\infty$) between open and closed states. Note that near the closing transition ($1 \le \Lambda \le 8$), there is significant variability in the final capillary rise, $\ell_I^\infty$. This variability, which is also visible in the second image of Fig. 2a, occurs within the same sheet (i.e. for the same $\Lambda$) and is not taken into account by our model, which assumes identical behaviour for each groove. We attribute this variability to small differences in groove sizes, which can substantially affect the equilibrium height. Additionally, these size differences can result in uneven forces acting on either side of the wall, potentially causing bending and symmetry-breaking events. Such effects, commonly observed in elastocapillary phenomena[24,38,39], greatly amplify variations in the groove cross-sections and, consequently, the equilibrium capillary rise height. Although the precise geometry of the groove of a hummingbird tongue is quite different, the dimensionless number $\Lambda$ may be estimated thanks to data from the literature[31,33] ($h \approx d \approx 150\,\mu\text{m}$, $e \approx 25\,\mu\text{m}$, $E \approx 300\,\text{kPa}$, $L \approx 1\,\text{cm}$, $\gamma \approx 0.06\,\text{N/m}$, $\rho \approx 1000\,\text{kg/m}^3$) and yields $\Lambda \approx 10^{-2}$, which is well inside the closing regime.

One can now compute the amount of liquid captured by the grooved sheet depending on its flexibility (Fig. 2e). For the rigid and flat case, the captured volume simply reads $\mathcal{V}_R = nhd\ell_l^{\infty, \text{open}}$, where $n \approx 2\pi h/d$ is the number of grooves. When the structure is flexible, however, it closes and the grooves become triangular, leading to a volume captured in the grooves $\mathcal{V}_F = nhd\ell_l^{\infty, \text{closed}}/2 = \mathcal{V}_R(1 + \bar{d}/2)$. Moreover, as the whole sheet curls and closes, an additional volume is trapped in the core of the newly formed tube of radius $R_{int} = hw/d$, leading to an additional volume $\mathcal{V}_C = \pi R_{int}^2 \ell_{II}^\infty = V_R w/2h$ and hence a total captured volume $\mathcal{V}_C + \mathcal{V}_F$ increase of a factor $1 + (w+d)/2h$ with

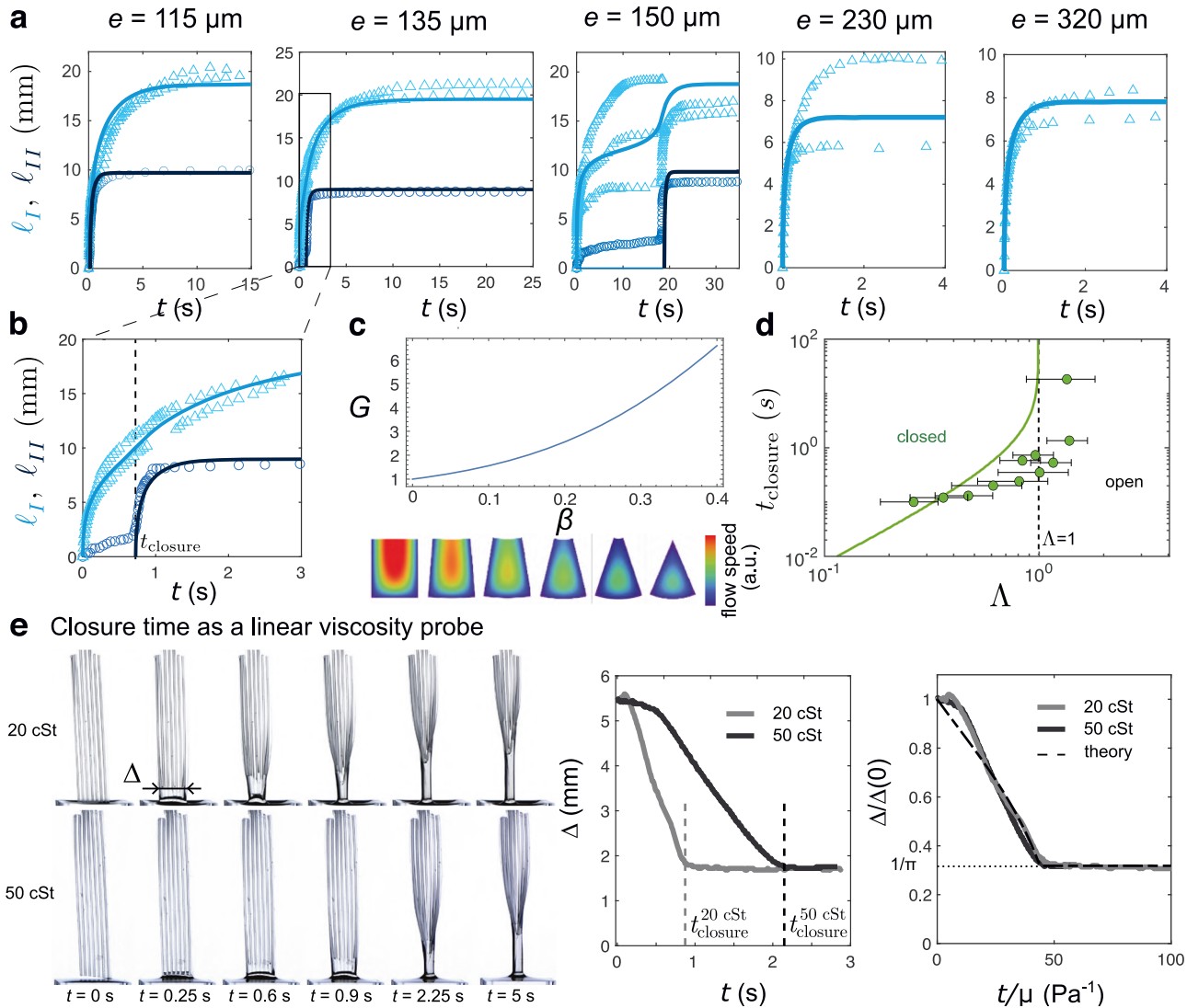

**Fig. 3 | Dynamics of the fluid-structure interaction. a** Evolution of the first capillary rises $\ell_I$ in two to three different grooves (triangles) and of the second capillary rise $\ell_{II}$ in the tube (circle), corresponding to the 5 different cases illustrated in Fig. 2a. The solid lines correspond to the theory. **b** Zoom on the first instant of the dynamics of capillary rise for a membrane of thickness $e = 135 \, \mu m$. **c** Evolution of the dimensionless hydraulic resistance of the grooves, $G$, as a function of $\beta$ (top) and illustration of the flow speed in the section of a closing tube (bottom). **d** Time needed for the structure to close after contact with the liquid bath, $t_{\text{closure}}$, as a function of the transition parameter. This time diverges at the transition. Solid lines and shaded area correspond to the theory with the uncertainty on the geometrical parameters. Error bars are estimated using the theory of error propagation. **e** The closure time of the device acts as a linear probe for the viscosity of the liquid. The evolution of the apparent width $\Delta$ of the device may be collapsed by rescaling the time by the viscosity ($w = 300 \pm 30 \, \mu m$, $d = 500 \pm 30 \, \mu m$, $h = 600 \pm 40 \, \mu m$, $e = 110 \pm 10 \, \mu m$, $L = 25.4 \pm 0.2 \, mm$). The dashed line correspond to the model for $h = 600 \, \mu m$, $w = 300 \, \mu m$, $d = 500 \, \mu m$, $e = 104 \, \mu m$, $L = 25 \, mm$. Source data are provided as a Source Data file.

respect to the rigid flat case, in good agreement with experiments (Fig. 2e). The method for measuring the volume of liquid captured is described in the Supplementary Information. In Fig. 2e, the transition once again takes place at $\Lambda = 1$, confirming the validity of the closure criterion.

**Dynamics of the sequential capillary rise**

Having rationalised under which condition the first capillary rise may induce a closure of the structure, and hence, a second capillary rise, we now aim to describe the dynamics of this intricate fluid-structure interaction problem. Some examples of experimental capillary rise dynamics are shown in Fig. 3a, b where light blue triangles (dark blue circles) show the temporal evolution of the first (second) capillary rise height. Following the seminal reasoning of Washburn, we balance the driving capillary force, $\mathbf{F}_\gamma$, with the resisting gravitational and viscous forces, $\mathbf{F}_g$ and $\mathbf{F}_\mu$, inside each groove of varying geometry due to elastic

deformation. Note that $\mathbf{F}_\gamma$ and $\mathbf{F}_g$ have already been derived in the previous section.

The derivation of the viscous friction induced by a Poiseuille-like flow in an open groove requires to solve the Stokes equations with proper boundary conditions, i.e. no-slip at liquid-solid interfaces and no-shear at liquid-air interfaces[13,14]. The derivation is detailed in the 'Methods' section 'Flow in a groove' and yields a viscous friction force of the form $\mathbf{F}_\mu = -\mu A^2(\beta, \bar{d})G(\beta, \bar{d})d^{-4}\ell_I(d\ell_I/dt)\mathbf{e_z}$, where $\mu$ is the liquid viscosity and $G$ is a dimensionless hydraulic resistance varying with $\beta$ and $\bar{d}$ (see Supplementary Fig. 2). The closing of the groove, governed by $\beta$, engenders a strong increase of the hydraulic resistance, as highlighted in Fig. 3c. Balancing the three forces yields a nonlinear differential equation for $\ell_I$. Yet, as the system has two unknowns ($\ell_I$ and $\beta$) we need a second equation to close the problem. This additional equation is given by the mechanical torque balance in the sheet, given by Eq. (3). To summarise, we obtain the following

system of equation for $\beta$ and $\ell_I$:

$$2\gamma(1+\beta)h - \rho g A(\beta,\bar{d})\ell_I - \mu A^2(\beta,\bar{d})G(\beta,\bar{d})d^{-4}\ell_I\frac{d\ell_I}{dt} = 0 \qquad (6)$$

$$B\frac{\beta}{2d} - \left(\frac{1+\beta}{2[\bar{d}-\beta]}+1\right)\gamma h\ell_I = 0, \qquad (7)$$

together with the initial condition $\ell_{I(0)} = 0$. Note that only one initial condition is needed since the second equation is algebraic. Solving numerically this nonlinear system of equations yields the solid light blue lines in Fig. 3a, b (see 'Methods' section 'Fluid-structure interaction and dimensionless equations' for more details), showing excellent agreement with the experiments within the intrinsic experimental uncertainties. We again attribute the variability in the final capillary height within each structure to small variations in groove dimensions. For sheets composed of $n \approx 2\pi/\bar{d}$ grooves, contact between neighbouring walls corresponds to the deformation of the whole structure into a tube. If the number of grooves is smaller (rsp. larger) than $n$, the sheet curls into a portion of a tube (rsp. an overlapping tube, see Supplementary Information). When the number of groove is sufficient to close the device, a second capillary rise, $\ell_{II}(t)$, takes place. Because self-contact induces a large increase of the bending stiffness of the sheet, this second capillary rise does not induce additional elastic deformation in the structure, and it follows the classical dynamics in a rigid tube[8,9]. Figure 3a, b shows excellent agreement between the model and the experiments for $\ell_{II}(t)$ (see also 'Methods' section 'Second capillary rise dynamics' and Supplementary Video 2). Varying the thickness $e$ and the length $L$ of the sheet, the time the structure takes to fully close after having being put in contact with the liquid bath, $t_{closure}$, may vary and can be theoretically computed as a function of $\Lambda$ (see 'Methods' sections 'Stationary solution and condition for groove closure' and 'Time needed for groove closure' and Supplementary Fig. 4). Fig. 3d shows that the closure time increases with $\Lambda$ and diverges at the transition (Supplementary Video 3). This increase is well captured by our model: a higher critical capillary rise is indeed needed to have enough force acting on the walls to bend the sheet when $\Lambda$ approaches 1, and reaching a higher elevation takes more time. The experimental critical value of $\Lambda$ at which $t_{closure}$ diverges is found to be slightly above 1 ($\approx 1.4$). This difference is due to a slight overestimation of the bending energy of the structure in the model, which must be bent by an angle $\beta$ over its entire length, $L$, to close, whereas, in the experiments, it closes when it is bent over a portion of its entire length (see Fig. 1b at $t = 1$ s). At lower values of $\Lambda$, the slight discrepancy between theory and experiments is due to the small value of $t_{closure}$ for which inertial effects of the liquid become significant and are not taken into account in the model (see Supplementary Information). Additionally, the closure time linearly increases with the viscosity of the liquid (Fig. 3e and Supplementary Video 4), provided the wetting properties remain constant. Hence, the sheet can be used as a simple viscosity probe by simply measuring the closure time.

### Full dipping
We now turn to the case where the liquid is abundant, and the device is dynamically fully dipped into a bath. The key advantage of the flexibility of the device, but also of the hummingbird tongue, resides in the amount of fluid captured in the process (see Supplementary Videos 5 and 6). Indeed, when the sheet is immersed in and then removed from a liquid bath at a speed $V$, the grooves are immediately filled with liquid during the protrusion phase (Fig. 4a, c). During the retraction phase, surface tension induces the closure of the sheet into a tube, capturing the liquid which slowly drains from the closed grooves (Fig. 4a). When a rigid structure with the same closed geometry (Fig. 4b) is dipped into the bath, the penetration of the liquid

inside the structure is much slower, as it has to flow through the cavity of the cylinder that has a large hydraulic resistance because of its small size (Fig. 4b, c). The driving force is the difference in hydrostatic pressure, that scales as $\rho g L$, whereas viscous shear, scaling as $\mu v L/R_{int}^2$—where v is the typical penetrating front velocity—resists the flow. Balancing both driving and resisting terms yields the typical penetrating speed $v \sim \rho g R_{int}^2/\mu$ that needs to be compared with the imposed dipping velocity $V$. For very slow dipping $V \ll v$, i.e. when $\mu V/[\rho g R_{int}^2] \ll 1$, the dipping is quasistatic, the fluid has the time to fully penetrate during the protrusion phase and then to drain during the retraction phase, leading to the same amount of liquid captured as in the static capillary rise (Fig. 4d). Both rigid and flexible structures thus capture the same amount of fluid. For very fast dipping however, i.e. when $\mu V/[\rho g R_{int}^2] \gg 1$, the liquid does not have the time to penetrate inside the rigid closed structure, leading to a strong decrease of the captured volume, whereas, for the flexible ones, it penetrates the grooves with negligible friction during protrusion and barely drains during retraction, thus exhibiting a strong increase in the amount of liquid trapped (Fig. 4c, d). The model, shown by the solid (rigid structures) and dashed lines (flexible structure) in Fig. 4c, d, consists of a simple Lucas-Washburn law with a hydrostatic pressure at the groove entrance $p(z = 0, t)$ varying with the dipping depth $D(t)$, $p(z = 0, t) = \rho g D(t)$, where $D(t) = Vt$ during immersion and $D(t) = L - Vt$ during retraction (see 'Methods' section 'Full immersion of the structure into the liquid bath'). Applying biological data of the hummingbird tongue from the literature[31,33] ($R \approx 0.1$–$0.2$ mm, $V \approx 0.1$–$0.4$ m/s and $\mu \approx 10^{-3}$–$10^{-1}$ Pa.s) to our model, we deduce that hummingbird tongues are precisely in the regime where flexibility strongly promotes fluid capture compared to its rigid counterpart (purple region in Fig. 4d).

## Discussion
This article demonstrates the efficacy for fluid capture of a flexible hierarchical structure composed of a set of vertical grooves that individually mimic the behaviour of a hummingbird's tongue. The intricate dynamics of concomitant fluid capture and structure deformation of the metatongue device, effectively described by a comprehensive analytical model that couples fluid flow at low Reynolds numbers with linear slender elasticity, occurs if two essential conditions are fulfilled: first, the width of the groove, $w$, must be smaller than the capillary length, $\ell_c$, for capillary rise to occur, and second, the capillary forces must surpass the mechanical resistance of the lamellae beneath the grooves (i.e. $\Lambda < 1$) to observe the structure deformation. No limitation exists on the whole sheet width, which can be much larger than the capillary length while still deformed by capillary forces. This is because the capillary actuation is local and occurs at the scale of the grooves, satisfying $w < \ell_c$. From a mechanical perspective, similarly to the supporting rod in hummingbird tongues, the ribs contribute to the structural rigidity of the sheet in the dipping direction, thereby limiting viscosity-induced buckling of the structure as it enters the liquid bath. Furthermore, this configuration notably extends the curvature persistence length of the entire structure, a topic of interest for future research. Comparative analysis of liquid capture by flexible structures versus rigid open or closed structures with identical texture geometry reveals that open rigid structures capture less liquid than flexible or closed rigid structures and that, when fully dipped into a liquid, fluid capture is notably enhanced and more rapid with flexible structures than with closed rigid ones. Such comparison shows that evolution equipped hummingbirds with an ideal and versatile tool for liquid capture, that efficiently operates both when nectar is scarce (edge dipping and capillary rise) or abundant (full dipping and fluid trapping). Compared to the hummingbird tongue, our device goes one step further, as its hierarchical design, leading to the closure of the whole sheet into a tube, enables the capture of an additional volume of liquid. Moreover, in contrast to rigid closed tubes, the flexible

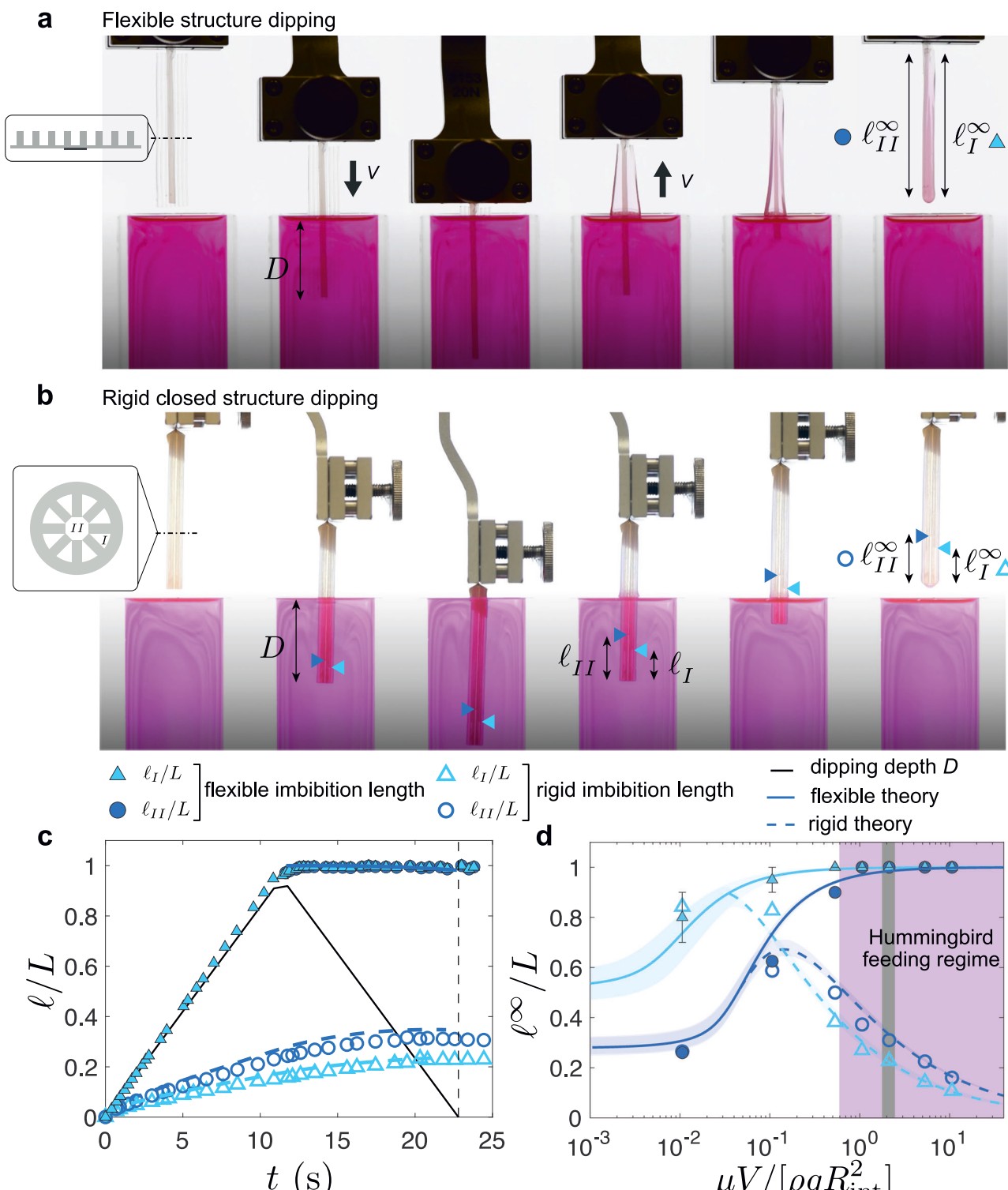

**Fig. 4 | Elasticity-enhanced fluid capture during dipping. a** Snapshots of a flexible grooved sheet as it is dipped and pulled out from a bath (silicone oil V1000, $\mu = 0.96$ Pa.s, $\rho = 960$ kg.m$^{-3}$, $\gamma = 0.021$ N/m) at constant speed $V = 200$ mm/min. Sheet geometry: $L = 40 \pm 0.2$ mm, $h = 800 \pm 20$ μm, $w = 360 \pm 20$ μm, $d = 700 \pm 50$ μm, $e = 135 \pm 10$ μm. **b** Dipping of a rigid closed structure with the same geometry and parameters. The position of the imbibing liquid fronts both inside the peripheral triangles $\ell_I$ and the central tube $\ell_{II}$ are highlighted by blue arrows. **c** Rescaled imbibition length $\ell/L$ as a function of time during the dipping cycle shown in (**a**, **b**); empty (rsp. filled) symbols correspond to the rigid (rsp. flexible)

case. The black line indicates the dipping depth as a function of time. Dashed lines correspond to the impregnation theory (see 'Methods' section `Full immersion of the structure into the liquid bath' for more details). **d** Final imbibition length $\ell^\infty/L$ (indicated by the vertical dashed line in **c**) as a function of the dimensionless number $\mu V/[\rho g R_{\mathrm{int}}^2]$. Error bars represent the variability of capillary rise across different grooves. Solid (rsp. dashed) lines correspond to the flexible (rsp. rigid) imbibition theory. Grey region indicates the data corresponding to (**c**). The hummingbird feeding regime is highlighted in purple. Source data are provided as a Source Data file.

**a**   Surface treatment: aqueous sampling for blood testing

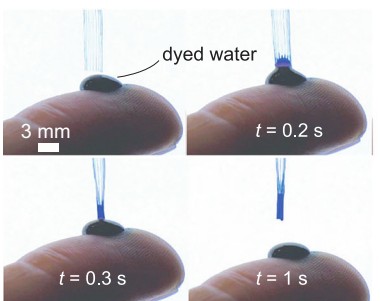
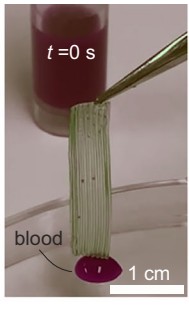
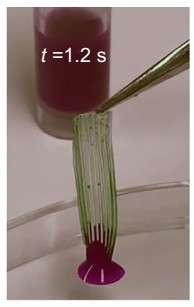
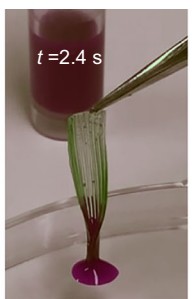
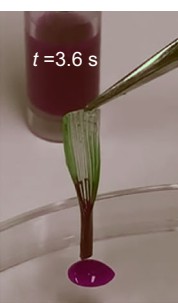

**b**   Groove functionalisation: sampling and aliquoting for efficient parallel testing

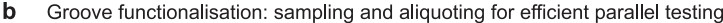
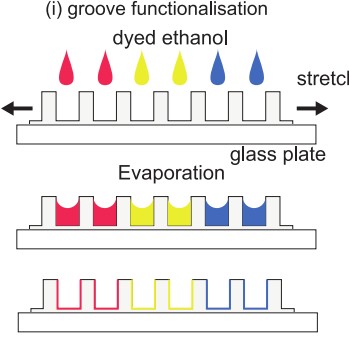
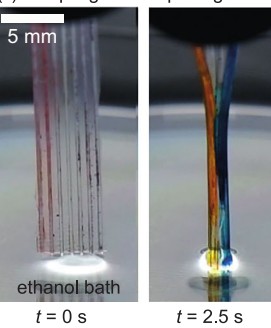
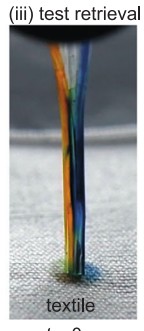
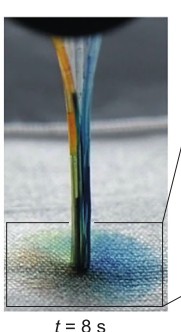
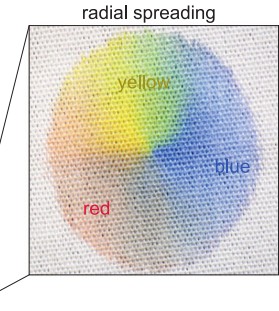

**Fig. 5 | Proof-of-concept towards all-in-one fluidic testing devices. a** Thanks to its simple geometry, the device may be easily scaled down and plasma treated to swiftly capture a small fraction of a water drop. The initial open state enables a fast flow through the grooves, and the closing of the structure impedes evaporation and unwanted exchanges with the environment ($h = w = d = 300 \pm 30\ \mu m$, $e = 60 \pm 5\ \mu m$). Additional treatment with serum albumin enables long-lasting hydrophilicity and the capture of blood (right panel, see 'Methods' section 'Surface treatment of the elastomer'). **b** Each groove may be additionally functionalised with specific markers (here, dye) using e.g. evaporation. Stretching the device, and using the strong adhesion on glass, the grooves may be made transitorily larger to ease this step. When dipped in a fluid (here, ethanol), the device swiftly samples and aliquots the liquid in each groove, where it efficiently reacts with the specific marker thanks to the triangular groove cross-section that ensures a large surface-to-volume ratio. The device is then put in contact with an absorbent textile, inducing the radial spreading of the captured liquid, enabling a direct reading of the role of the different markers. Source data are provided as a Source Data file.

device possesses the remarkable ability to be initially flat, enabling straightforward storage, and can be reopened through simple manual manipulations (see Supplementary Video 7), simplifying the retrieval of the captured fluid. Natural extensions of our work can exploit the variation of the geometries of the textures and the ability to design structures that curl into closed tubes with diverse and potentially more efficient cross-sectional geometries[36], or on the design of structures with flexible walls that may collapse[38] (see Supplementary Video 8).

Combining capillary suction and structure deformation thus opens new routes in passive capillary microfluidics[40,41], especially for biological assays[42] and medical diagnostics[43] like blood testing[44], as it offers the opportunity to concomitantly collect and aliquot the sample in a fast and efficient manner. As a first step towards such applications, we show in Fig. 5a that the structure can be scaled down (left panel) and plasma treated to sample and aliquot few microliters of an aqueous solution (see Supplementary Video 9). As plasma exposure only gives temporary hydrophilicity to the structure, an additional treatment is performed by filling the grooves with a solution of serum albumin, that is dried in an oven (see 'Methods' 'Surface treatment of the elastomer' for more details). In Fig. 5a, right panel and Supplementary Video 10, blood is sampled in such treated flexible silicone device. However, while serum albumin-treated silicone retains hydrophilic properties for at least 1 day, the precise duration of this effect was not quantified. Consequently, ensuring long-term surface hydrophilicity may represent a significant challenge for the application of such devices in blood diagnostics and could require a change in the device material. Each individual groove can be additionally

functionalised with specific markers. As a first proof of concept, we deposit dyed ethanol (red, blue, yellow), in grooves (Fig. 5b, left panel). This process is facilitated by the flexibility of the device, that can be stretched and adheres well on a glass slide. After evaporation, our model pre-coated reagents (here, dyes crystals) are present in the aliquots (the grooves). When put in contact to a wetting liquid (here, ethanol), the reagents are quickly released in the solution (Fig. 5b, central panel and Supplementary Video 11). In addition, contacting the tip of the imbibed structure to an absorbent substrate allows for a direct reading of the reaction (here, only diffusion—see Fig. 5b, right panel and Supplementary Video 11). We believe that replacing the dye with dehydrated antibodies in the grooves, which requires additional work beyond the scope of this study, should allow for rapid blood tests using only a fraction of a drop volume instead of the three droplets needed, for example, in a Serafol® ABO+D card used to confirm patient identity immediately before a blood transfusion. The reading could also be performed after the structure is reopened and the groove in which blood agglutination has occurred is examined. Finally, as the closure time is related to the liquid viscosity (Fig. 3e), it could rapidly inform on the viscosity of the biological fluid of interest, that can be directly correlated to some critical properties, such as the haematocrit level in blood[45–47]. As a whole, our device has the potential to become an all-in-one fluidic testing platform and may offer an alternative approach to existing testing techniques, such as centrifugal microfluidics[48] for at least blood testing, as it is passive, fast, affordable, and requires a small volume sample. These attributes are also compatible with those of paper-based microfluidics making medical diagnostics accessible for developing countries[49].

## Methods

### Elastomer preparation

The structures are made of platinum-catalysed silicone rubber (Ecoflex 00-50 from Smooth-On) and are fabricated by mixing a prepolymer base and a curing catalyst in a 1:1 weight ratio. Just after mixing, when the melt is still liquid, it is poured on a 3D-printed mould (printed with an Ultimaker S5) and spin-coated (SCS 6800 SpinCoater Series) at various rotation speeds that control the sheet thickness. The polymer melt gradually cures over time leading to an elastic solid. At room temperature curing time is 2 h. After demoulding the sheets, they are placed in a bath of V10 silicone oil (Merck, reference number 378321) until they are completely swollen (typically overnight), leading to a uniform increase in lengths of 30%. This step enables us to use silicone oil as a completely wetting liquid (contact angle $\theta_Y = 0$), without inducing any additional swelling of the sheet. The Young's modulus of the swollen elastomer is 57 kPa measured by mean of a traction machine (ZwickiLine Z0.5TH from Zwick) assuming material incompressibility (i.e. a Poisson ratio $\nu = 0.5$).

### Surface treatment of the elastomer

When using an aqueous solution as the wetting liquid, it is essential to treat the elastomer's surface to ensure proper wettability. This process involves plasma treatment of the unswollen sample, which temporarily makes the surface hydrophilic for a few hours. To further enhance and prolong hydrophilicity, a 2 wt% serum albumin solution can be poured into the grooves. The sample is then placed in an oven at 40 °C for 1 h to evaporate the serum albumin solution. Once dried, the treated sample becomes hydrophilic and is ready for use in capturing aqueous solutions, such as blood.

### Properties of the fluid-structure couples used in the article

Table 1 resumes the range of all geometrical parameters of the samples, the physical and chemical properties of the wetting liquids and the mechanical properties of the elastomers used in the article.

### Experimental apparatus

A schematic of our experiments is shown in Supplementary Fig. 1.

The structured sheets are hanged by mean of a traction machine (ZwickiLine 0.5 kN from Zwick) above a liquid bath filled with silicone oil (V10, V20, V50 or V1000 from Merck) with the grooves oriented vertically. Note that unless specified the wetting liquid is a silicone oil V10. The structure is then either slightly dipped in the liquid reservoir or fully immersed in and withdrawn from the bath at a controlled velocity. While capillary rise occurs, the dynamics is recorded using a camera (Nikon D850) at a frame rate of 100 fps or 120 fps. The height of the first and second capillary rise are then extracted from the images by standard image processing using ImageJ and MATLAB. In the experiments involving the full immersion of the flexible sheet into the liquid bath, a lamella of Mylar of width comparable to the ribs width $d$, and of same length $L$ as the sheet, is fixed at the back of the structure to increase its structural rigidity in the dipping direction in order to prevent viscosity-induced buckling of the structures.

### Modelling approach

We start by presenting the general ideas of the model before giving the details in the next sections. To model the fluid-structure problem occurring when a fluid rises in a deformable grooved sheet, we consider the flow inside each groove, where the closing angle $\beta$ is the only degree of freedom in the deformation. The computation of the flow is based on the Washburn approach which assumes that the unidirectional flow is driven by the capillary pressure gradient ($\Delta p$) caused by the free surface, while viscous and gravity forces oppose the flow[7]. For the sake of simplicity, we neglect the inertial terms in the momentum equation, as they are significant only for the very early dynamics of the capillary rise and are thus irrelevant for our fluid-structure interaction study (see Supplementary Information). Under these conditions, the flow rate $Q$ has a Poiseuille-like expression

$$Q = \frac{1}{\mu}\left[\frac{\Delta p}{\ell_l} - \rho g\right] d^4 \, G^{-1}(\beta, \bar{d}), \qquad (8)$$

where $\bar{d} = d/h$, $G$ is a function taking into account the geometry of the groove, and $\ell_l$ is the liquid height in a single groove (see Supplementary Fig. 2a).

On the other hand, the flow rate is also equal to $Q = A(\beta, \bar{d})d\ell_l/dt$, where $A$ is the area of an horizontal cross-section of the groove. Equating these two expressions gives an evolution equation for $\ell_l(t)$ once $\Delta p$ and $G$ are known. This equation involves $\beta$ which varies as the fluid rises. A second equation is thus needed to close the system. It is obtained from the balance of torques due to the bending of the sheet, the capillary pressure and the capillary force at the triple line. The fluid-structure problem is then solved by computing numerically the solution of these two coupled equations describing the simultaneous capillary rise inside the grooves, i.e. $\ell_l(t)$, and the bending of the sheet, i.e. $\beta(t)$.

### Geometry of a groove and of the air-liquid interfaces

A groove is composed of two rigid walls of height $h$ attached to a flexible sheet of width $d$ and having both a length $L$, see Supplementary Fig. 2a. When the fluid rises inside the groove, the walls move toward each others and their clamped ends rotate by an angle $\beta$ with respect to their initial position so that the distance between their free ends decrease. During this process, the flexible sheet bends. For simplicity, we assume that it bends along an arc of circle and that the walls stay perpendicular to the sheet during their rotation. The sheet and the walls form then a circular sector with a central angle $2\beta$ so that the radius of curvature of the sheet is given by (Supplementary Fig. 2b)

$$R = \frac{d}{2\beta}. \qquad (9)$$

We also assume for simplicity that the vertical air-liquid interface $A'$ joining the wall free ends is bent along an arc of circle of radius $R - h$ with the same central angle $2\beta$ (Supplementary Fig. 2a). The angle $\theta$ between the liquid and the walls at the wedge is thus constant and equal to $\pi/2$ in this case. Note that this angle is compatible with a zero wetting contact angle (i.e. a perfectly wetting liquid). In fact, the wedge of the ribs is a singular point where the angle of the liquid-solid

**Table 1 | Range of variation of the geometrical, chemical and physical properties of the liquids and the membranes used in the paper**

| Geometry | | | | | Liquids properties | | | Membranes properties | | Transition parameter |
|---|---|---|---|---|---|---|---|---|---|---|
| $L$ (mm) | $e$ (µm) | $w$ (µm) | $d$ (µm) | $h$ (µm) | $\mu$ (mPa.s) | $\gamma$ (mN/m) | $\rho$ (kg/m³) | $E$ (kPa) | $\nu$ | $\Lambda$ |
| 22–50 | 60–320 | 300–400 | 300–700 | 300–850 | 1–960 | 21–72 | 785–1000 | 57–90 | 0.5 | 0.022–9.7 |

interface can range from 0 to $\pi/2$ if the liquid perfectly wets the surface. Here, an angle of $\pi/2$ corresponds to a zero contact angle relative to the top of the walls. At any vertical elevation $z$, the length $d'$ of this interface is thus

$$d' = 2\beta(R - h) = d - 2h\beta, \quad 0 \le \beta \le d/(2h). \tag{10}$$

When $\beta = d/(2h)$, free ends of the walls are in contact and $d' = 0$. The area of the vertical air-liquid interface is thus given by

$$A'(\beta, \bar{d}) = d' \, \ell_l. \tag{11}$$

Neglecting the curvature of the horizontal air-liquid interface $A$, its area coincides with the area $A$ of a horizontal cross-section of a groove given by the difference between the area of two circular sectors of radius $R$ and $R-h$ with the same central angle $2\beta$:

$$A(\beta, \bar{d}) = \beta R^2 - \beta(R - h)^2 = h^2 (\bar{d} - \beta) \equiv h^2 \, \bar{A}(\beta, \bar{d}), \tag{12}$$

where $\bar{d} = d/h$.

In this simplified geometry, the centres of curvature of both the sheet and the vertical air-liquid interface $A'$ are located on the same vertical line where the $z$-axis of a cylindrical system of coordinates will be placed to compute the flow inside the groove in the next section. Doing so, the boundaries of a groove are described by coordinate lines, allowing to separate variables when solving the Stokes equations.

## Flow in a groove

The flow inside a groove is described in cylindrical coordinates ($\mathbf{e}_r$, $\mathbf{e}_\phi$, $\mathbf{e}_z$) whose origin is located at the bottom of the groove on the centre of curvature of the flexible sheet, see Supplementary Fig. 2b. The groove is thus described by $(r, \phi, z)$ with $R-h \le r \le R$, $-\beta \le \phi \le \beta$ and $0 \le z \le \ell_l$ and its cross-sectional area is given by Eq. (12). We assume the flow to be unidirectional along the $z$-direction, so that the $r$ and $\phi$ component of the velocity field $\mathbf{u}$ are vanishing ($u_r = u_\phi = 0$), and the fluid to be an incompressible Newtonian fluid of density $\rho$ and viscosity $\mu$. In this case, mass conservation imposes the following form for $\mathbf{u}$:

$$\frac{\partial u_z}{\partial z} = 0 \Rightarrow \mathbf{u} = u_z(r, \phi)\mathbf{e}_z. \tag{13}$$

The Navier-Stokes equations in cylindrical coordinates simplifies to:

$$\rho \frac{\partial u_z}{\partial t} = -\rho g - \frac{\partial p}{\partial z} + \mu \nabla^2 u_z, \quad \frac{\partial p}{\partial r} = \frac{\partial p}{\partial \phi} = 0, \tag{14}$$

where $g$ is the gravitation acceleration, i.e. $\mathbf{g} = -g\mathbf{e}_z$. Because $u_z$ does not depend on $z$, taking the $z$-derivative of Eq. (14) leads to $\partial_z(\partial_z p) = 0$ so that:

$$p(z) = p_{\text{atm}} - \frac{\Delta p}{\ell_l} z, \tag{15}$$

where $p_{\text{atm}}$ is the atmospheric pressure, $\Delta p$ is the pressure difference between the entry ($z = 0$) and the exit of the groove ($z = \ell_l$) and $\ell_l$ is the length of the groove filled by the liquid. Searching for stationary solution and writing the Laplacian operator in cylindrical coordinates, the first of Eq. (14) simplifies to:

$$\frac{\partial^2 u_z}{\partial r^2} + \frac{1}{r}\frac{\partial u_z}{\partial r} + \frac{1}{r^2}\frac{\partial^2 u_z}{\partial \phi^2} + \frac{1}{\mu}\left[\frac{\Delta p}{\ell_l} - \rho g\right] = 0. \tag{16}$$

We assume a usual non-slip boundary condition at the liquid-solid interface. This assumption is justified even if the solid has been swollen using the same liquid (silicone oil V10) in most of the experiments.

Indeed, if a non-zero slip length is present, it would likely correspond to the size of the polymer mesh or the polysiloxane chains of the silicone oil. Both are of the order of few tens of nanometres, which is negligible compared to the other characteristic lengths involved in the system. We thus assume that the velocity vanishes along walls:

$$u_z(r, \phi = \pm \beta) = 0. \tag{17}$$

The following quadratic ansatz is commonly used to satisfy those two boundary conditions[50,51]:

$$u_z(r, \phi) = v_z(r)\,\varphi(\phi) = v_z(r) \, \frac{3(\beta^2 - \phi^2)}{2\beta^2}. \tag{18}$$

Substituting Eq. (18) into Eq. (16), we obtain

$$\varphi \frac{d^2 v_z}{dr^2} + \frac{\varphi}{r}\frac{dv_z}{dr} + \frac{v_z}{r^2}\frac{d^2\varphi}{d\phi^2} + \frac{1}{\mu}\left[\frac{\Delta p}{\ell_l} - \rho g\right] = 0. \tag{19}$$

We now compute the mean flow along $z$ by taking the average along the $\phi$ direction. Noting that

$$\frac{1}{2\beta}\int_{-\beta}^{\beta} \varphi(\phi)\, d\phi = 1, \quad \frac{1}{2\beta}\int_{-\beta}^{\beta}\frac{d^2\varphi}{d\phi^2}\, d\phi = -\frac{3}{\beta^2}, \tag{20}$$

The $\phi$-average of Eq. (19) reads:

$$\frac{d^2 v_z}{dr^2} + \frac{1}{r}\frac{dv_z}{dr} - \frac{3}{\beta^2}\frac{v_z}{r^2} + \frac{1}{\mu}\left[\frac{\Delta p}{\ell_l} - \rho g\right] = 0. \tag{21}$$

Thanks to this ansatz, variables have been separated, and we are left with an ordinary second-order differential Eq. (21) for $v_z$, which requires two boundary conditions to be solved. The no-slip condition at the flexible sheet gives a first boundary condition:

$$v_z(r = R) = 0, \tag{22}$$

where $R$ given by Eq. (9), is the radius of curvature of the sheet. At the air-liquid interface, that we assume to be at a constant $r = R - h$ as we did to compute $A$ [Eq. 12], we impose a no-stress boundary condition:

$$\frac{\partial v_z}{\partial r}\left(r = R_c = R - h\right) = 0. \tag{23}$$

The solution of Eq. (21) together with the boundary conditions (22) and (23) reads:

$$v_z(\bar{r}) = \frac{d^2}{\mu}\left[\frac{\Delta p}{\ell_l} - \rho g\right] v(\bar{r}, k, \bar{R}_c), \tag{24a}$$

$$v = c_1 \bar{r}^k + c_2 \bar{r}^{-k} + \frac{k^2 \bar{r}^2}{12(k^2 - 4)}, \tag{24b}$$

$$c_1 = -\frac{k(k + 2\bar{R}_c^{k+2})}{12(k^2 - 4)(1 + \bar{R}_c^{2k})}, \tag{24c}$$

$$c_2 = \frac{k\bar{R}_c^k(2\bar{R}_c^2 - k\bar{R}_c^k)}{12(k^2 - 4)(1 + \bar{R}_c^{2k})}, \tag{24d}$$

where $\bar{r} = r/R$, $\bar{R}_c = R_c/R = 1 - h/R$ and $k = \sqrt{3}/\beta$. This solution is valid provided $k \ne 2$. An exact solution also exists for $k = 2$ but we do not write it here because we assume $d/h \lesssim 1$ so that $k > 2$ because $\beta \le d/(2h)$.

Note that, using Eq. (9), we also have $\bar{R}_c = 1 - 2\beta/\bar{d}$ with $\bar{d} = d/h$. In addition, $r$ varies between $R-h$ and $R$ so that $x = (R-r)/h = R/h(1-\bar{r})$ varies between 0 when $r = R$ and 1 when $r = R-h$. Therefore, the dimensionless mean vertical velocity v can be plotted as a function of $x$ with $\bar{d}$ and $\beta$ as parameters (instead of $k$ and $\bar{R}_c$) as shown in Supplementary Fig. 2c for $\bar{d} = 0.8$. Indeed, for a given groove, $\bar{d}$ is a constant and only $\beta$ varies during the fluid rise whereas both $k$ and $\bar{R}_c$ varies with $\beta$.

Finally, the complete velocity field is obtained from Eqs. (13) and (18):

$$\mathbf{u} = u_z(r,\phi)\,\mathbf{e}_z = 3\,v_z(r)\left[\frac{\beta^2 - \phi^2}{2\beta^2}\right]\mathbf{e}_z, \tag{25}$$

with v$_z$ given by Eq. (24). This velocity field is shown in Supplementary Fig. 2d for $\bar{d} = 0.8$ and several values of $\beta$.

To obtain an evolution equation for $\ell_l$ as a function of $\beta$ and $\bar{d}$, we now compute the flow rate $Q$

$$Q = \int_{-\beta}^{\beta}\int_{R_c}^{R} r\,u_z(r,\phi)\,d\phi\,dr = 2\beta R^2 \int_{\bar{R}_c}^{1} \bar{r}\,v_z(\bar{r})d\bar{r}. \tag{26}$$

where we used the change of variable $\bar{r} = r/R$ and Eq. (20) to compute the integral over $\phi$. Using Eqs. (9) and (24a), we have

$$Q = \frac{d^4}{\mu}\left[\frac{\Delta p}{\ell_l} - \rho g\right]\tilde{G}^{-1}(k,\bar{R}_c), \tag{27a}$$

$$\tilde{G}^{-1} = \frac{k}{2\sqrt{3}}\int_{\bar{R}_c}^{1}\bar{r}\,v(\bar{r},k,\bar{R}_c)d\bar{r}, \tag{27b}$$

where $\beta = \sqrt{3}/k$ have been used. The remaining integral can be computed leading to

$$\tilde{G}(k,\bar{R}_c) = \left[96\sqrt{3}\,k^{-2}\,(k^2-4)^2\,(1+\bar{R}_c^{2k})\right] \times$$
$$\left[k(k-2)^2 - (k-4)(k+2)^2\,\bar{R}_c^4 + k(k+2)^2\,\bar{R}_c^{2k} \right.$$
$$\left. -32k\bar{R}_c^{2+k} - (k+4)(k-2)^2\bar{R}_c^{4+2k}\right]^{-1}. \tag{28}$$

Again, the parameters of interest in our system are $\bar{d}$ and $\beta$ instead of $k$ and $\bar{R}_c$. We thus define

$$G(\beta,\bar{d}) = \tilde{G}(k(\beta),\bar{R}_c(\beta,\bar{d})), \tag{29a}$$

$$k = \frac{\sqrt{3}}{\beta}, \quad \bar{R}_c = 1 - \frac{2\beta}{\bar{d}}. \tag{29b}$$

Despite its apparent complexity, the behaviour of the function $G$ is quite simple in the regime of interest, as shown in Supplementary Fig. 2e, i.e. for $0 \leq \bar{d} \leq 1.1$ and for $0 \leq \beta \leq \bar{d}/2$ (upper bound corresponding to contact between neighbouring walls).

The evolution equation for $\ell_l$ is obtained by equating Eq. (27a) to its equivalent expression, namely $Q = A(\beta,\bar{d})d\ell_l/dt$ where $A$ is given by Eq. (12):

$$\frac{\mu}{d^4}A(\beta,\bar{d})\,G(\beta,\bar{d})\,\ell_l\,\dot{\ell}_l = \Delta p - \rho g\ell_l, \tag{30}$$

where $\dot{\ell}_l = d\ell_l/dt$. We thus need now to compute $\Delta p$, which is the remaining unknown quantity in this equation.

## Pressure difference in a groove

The pressure difference is due to the driving capillary force, which can be derived from the change $dF$ in the free energy of the system when the liquid moves from an height $\ell_l$ to an height $\ell_l + d\ell_l$ in the groove. The free energy reads as

$$F = \gamma A_{LV} + \gamma_{SL}A_{SL} + \gamma_{SV}A_{SV}, \tag{31a}$$

$$A_{LV} = A(\beta,\bar{d}) + A'(\beta,\bar{d}), \quad A_{SL} = (2h+d)\,\ell_l, \tag{31b}$$

$$A_{SV} = (2h+d)(L-\ell_l), \quad \gamma_{SV} - \gamma_{SL} = \gamma\cos\theta_Y, \tag{31c}$$

where $A_{LV}$, $A_{SL}$, and $A_{SV}$ are the areas of the liquid-vapour, solid-liquid, solid-vapour interfaces, respectively, $\gamma \equiv \gamma_{LV}$, $\gamma_{SL}$, and $\gamma_{SV}$ are the energy per unit area of the liquid-vapour, solid-liquid, solid-vapour interfaces, respectively, $A(\beta,\bar{d})$ is given by Eq. (12), $A'(\beta,\bar{d})$ is given by Eq. (11) and the last relation is the Young's equation. Considering $\theta_Y = 0$, the expression of $F$ becomes

$$F = \gamma\left[h^2(\bar{d}-\beta) - 2(1+\beta)h\,\ell_l\right] + \gamma_{SV}(2h+d)L. \tag{32}$$

The capillary pressure is then given by

$$\Delta p = -\frac{1}{A}\frac{dF}{d\ell_l} = \frac{2\gamma(1+\beta)h}{A(\beta,\bar{d})} = \frac{2\gamma(1+\beta)}{h(\bar{d}-\beta)}. \tag{33}$$

Substituting Eq. (33) in Eq. (30) and multiplying both side by $A$, we obtain the evolution equation for liquid height $\ell_l$ in the groove

$$\frac{\mu}{d^4}A^2(\beta,\bar{d})\,G(\beta,\bar{d})\,\ell_l\,\dot{\ell}_l = 2\gamma(1+\beta)h - \rho g A(\beta,\bar{d})\ell_l. \tag{34}$$

Note that this equation corresponds to the force balance mentioned in the article :

$$F_\mu + F_\gamma + F_g = 0, \tag{35}$$

where $F_\mu = -\mu A^2(\beta,\bar{d})G(\beta,\bar{d})\ell_l\dot{\ell}_l/d^4$ is the viscous resistive force, $F_\gamma = 2\gamma(1+\beta)h$ is the driving capillary force and $F_g = -\rho g A(\beta,\bar{d})\ell_l$ is the gravitational force. Equation (34) depends on $\beta$, which varies during the liquid rise, hence, we need a second equation relating $\ell_l$ and $\beta$, to get a closed system of equations. It is obtained from the torque balance.

## Torque balance

The flexible sheet in between neighbouring walls is subjected to a bending moment due to both the depression in the rising liquid column $M_p$ and the pulling contact line at the top of the walls $M_\gamma$. Hence, neglecting the sheet inertia, the elastic restoring torque $M_B$ is given at every instant $t$ by:

$$M_B = \frac{B}{R} = M_p + M_\gamma, \tag{36}$$

where $B = Ee^3L/[12(1-\nu^2)]$ is the bending modulus of the flexible sheet, $E$ and $\nu$ the Young modulus and the Poisson ratio of the material, respectively, and $R = d/(2\beta)$ the radius of curvature of the sheet. The moment due to depression in the groove reads

$$M_p = \frac{h^2}{2}\int_0^{\ell_l}(p_{atm} - p(z))dz = \frac{h^2\Delta p}{2\ell_l}\int_0^{\ell_l} z\,dz, \tag{37}$$

where $p(z)$ is given by Eq. (15) and $\Delta p$ by Eq. (33). We thus obtain

$$M_p = \frac{1+\beta}{2(\bar{d}-\beta)}\gamma h \ell_I. \tag{38}$$

The moment induced by the surface tension at the contact line is given by

$$M_\gamma = \gamma h \int_0^{\ell_I} \sin\theta \, dz = \gamma h \ell_I. \tag{39}$$

where $\theta = \pi/2$ is the angle between the vertical air-liquid interface and the walls.

The second equation relating $\beta$ to $\ell_I$ is obtained by substituting Eqs. (37) and (39) in Eq. (36)

$$\frac{2\beta B}{d} = \gamma h \ell_I \left[1 + \frac{1+\beta}{2(\bar{d}-\beta)}\right]. \tag{40}$$

## Fluid-structure interaction and dimensionless equations

The fluid-structure problem where a liquid rises in a deformable groove is addressed by solving the coupled Eqs. (34) and (40). For this purpose, using dimensionless equations is convenient. We rescale the variables as follows

$$\bar{d} = d/h, \quad \bar{A} = A/h^2, \quad \bar{\ell}_I = \ell_I/\ell_I^{\infty,\,\mathrm{open}}, \quad \bar{t} = t/\tau_I, \tag{41}$$

where

$$\ell_I^{\infty,\,\mathrm{open}} = \frac{2\ell_c^2}{d}, \quad \ell_c = \sqrt{\frac{\gamma}{\rho g}}, \tag{42}$$

are the stationary height of the liquid in a groove with $\beta = 0$ (as shown in 'Methods' section 'Stationary solution and condition for groove closure') and the capillary length, respectively, and where $\tau_I$ is given by

$$\tau_I = \frac{\mu}{\rho g}\frac{\ell_c^2 h^3}{d^6}. \tag{43}$$

With these change of variables, Eqs. (34) and (40) become

$$2(\bar{d}-\beta)^2 \, G(\beta,\bar{d}) \, \bar{\ell}_I \, \dot{\bar{\ell}}_I = (1+\beta) - \left[1 - \frac{\beta}{d}\right]\bar{\ell}_I, \tag{44a}$$

$$K\beta = \left[2 + \frac{1+\beta}{(\bar{d}-\beta)}\right]\frac{\bar{\ell}_I}{\bar{d}}, \tag{44b}$$

where $K = 2B/[\gamma d \ell_c^2]$ is the dimensionless stiffness of the sheet, and $G$ is defined by Eqs. (28) and (29). This nonlinear system of equations can be solved numerically using MATLAB or Mathematica with the initial condition $\bar{\ell}_I = 0$ at $\bar{t} = 0$.

In the next sections, we discuss the stationary solution, the short-time dynamics, the condition on $K$ for contact between neighbouring walls, and the time needed to achieve this contact.

## Stationary solution and condition for groove closure

The stationary solution is obtained by setting $\dot{\bar{\ell}}_I = 0$ in Eq. (44a). This substitution leads to

$$\bar{\ell}_I^\infty = \frac{(1+\beta)\bar{d}}{(\bar{d}-\beta)} \quad\Rightarrow\quad \ell_I^\infty = \frac{2\ell_c^2}{h}\frac{(1+\beta)}{\bar{d}-\beta}. \tag{45}$$

Therefore, $\ell_I^\infty$ varies between $\ell_I^{\infty,\,\mathrm{open}}$ and $\ell_I^{\infty,\,\mathrm{close}}$ when $\beta$ varies between 0 and $\bar{d}/2$ where

$$\ell_I^{\infty,\,\mathrm{open}} = \frac{2\ell_c^2}{d}, \quad \ell_I^{\infty,\,\mathrm{closed}} = \frac{2\ell_c^2}{d}(2+\bar{d}). \tag{46}$$

The value $\beta^\infty$ of $\beta$ corresponding to $\bar{\ell}_I = \bar{\ell}_I^\infty$ (and which fully determines $\bar{\ell}_I^\infty$ once substituted in Eq. (45)) is obtained by replacing $\bar{\ell}_I$ by $\bar{\ell}_I^\infty$ in Eq. (44b)

$$K\beta^\infty = \left[2 + \frac{1+\beta^\infty}{(\bar{d}-\beta^\infty)}\right]\frac{(1+\beta^\infty)}{(\bar{d}-\beta^\infty)} \equiv f(\beta^\infty, \bar{d}). \tag{47}$$

This equation is a third-order algebraic equation for $\beta^\infty$, which can be solved exactly. However, the expression of the solutions, which depends on $\bar{d}$ and $K$, is cumbersome and omitted here, as we focus on the case where the neighbouring walls are in contact, i.e. when $\beta = \bar{d}/2$, as discussed below.

Supplementary Fig. 3a shows a colour map and contour plot of $\beta^\infty$ as a function of $\bar{d}$ and $K$ and Supplementary Fig. 3b shows similar plots for $\bar{\ell}_I^\infty$ obtained by substituting $\beta^\infty$ in Eq. (45). For a given value of $\bar{d}$, say 0.8, the value of $\beta^\infty$ decreases to 0 and $\bar{\ell}_I^\infty$ tends to 1 as $K$ increases. Indeed, $K$ measures the stiffness of the sheet relative to capillary forces. When $K$ is large, the flexible sheet is stiff and barely bends ($\beta$ stays close to 0) and, consequently, $\bar{\ell}_I^\infty$ tends to the equilibrium height in an open groove, i.e. $\ell_I^\infty = \ell_I^{\infty,\,\mathrm{open}}$ and thus $\bar{\ell}_I^\infty = 1$, see Eq. (41).

As $K$ decreases, both $\beta^\infty$ and $\bar{\ell}_I^\infty$ increase until $K$ reaches a critical value $K_c$ below which there is no solution for Eq. (47) in the physical interval $0 \le \beta^\infty \le \bar{d}/2$. Indeed, the function $f(\beta^\infty, \bar{d})$ defined in Eq. (47) is singular at $\beta = \bar{d}$ and is thus composed of two branches. When $K < K_c$, Eq. (47) has only one real solution corresponding to the intersection between $K\beta$ and the second branch of $f$. The intersection is located at $\beta > \bar{d}$. Since $\beta$ cannot be larger than $\bar{d}/2$, we have $\beta = \bar{d}/2$ when $K < K_c$. We do not add a superscript $\infty$ to $\beta$ in this case since this extreme value of $\beta$ does not correspond to a mechanical equilibrium and is thus not a solution of Eq. (47).

The existence of a critical value of $K$ is illustrated in Supplementary Fig. 3c, where both functions $K\beta$ and $f(\beta,\bar{d})$ are plotted for $\bar{d} = 0.8$ and several values of $K$ as a function of $\beta$ within the physical interval $\beta \in [0, \bar{d}/2]$. The solution of Eq. (47) for a given $K$ corresponds to the intersection of these two functions of $\beta$, shown as star symbols. When $K < K_c$, no intersection occurs between the two function within the physical interval, and $\beta$ jumps from $\beta_c(\bar{d}) = \beta^\infty(\bar{d}, K_c(\bar{d}))$ to $\beta = \bar{d}/2$, because $\beta^\infty(\bar{d}, K < K_c)$ exceeds $\bar{d}/2$.

The critical value of $K$ is thus such that the linear function $K_c\beta$ is tangent to the function $f(\beta, \bar{d})$,

$$\left.\frac{\partial f}{\partial \beta}\right|_{\beta=\beta_c} = K_c. \tag{48}$$

Also, Eq. (47) gives the following relation for $K_c$:

$$K_c\beta_c = f(\beta_c, \bar{d}). \tag{49}$$

Eliminating $K_c$ from Eqs. (48) and (49), we get an equation for $\beta_c$

$$\left.\frac{\partial f}{\partial \beta}\right|_{\beta=\beta_c} = \frac{f(\beta_c, \bar{d})}{\beta_c} \tag{50}$$

where $f$ is given by Eq. (47). This equation is a third-order algebraic equation for $\beta_c$ which possesses only one solution in the physical interval for $\beta$, i.e. $[0, \bar{d}/2]$. This solution depends only on $\bar{d}$, in contrast to the solution of Eq. (47) for $\beta^\infty$ which depends on both $\bar{d}$ and $K$, and it

can be express in a rather compact form as follows

$$\beta_c(\bar{d}) = \bar{d} - (1+\bar{d})\left[\cos\sigma(\bar{d}) - \sqrt{3}\sin\sigma(\bar{d})\right] \approx \frac{\bar{d}}{3} - 0.028\,\bar{d}^2, \quad (51a)$$

$$\sigma(\bar{d}) = \frac{1}{3}\arctan\left(\sqrt{1+2\bar{d}}/\bar{d}\right). \quad (51b)$$

This function is shown in Supplementary Fig. 3d and is well approximated by a simple polynomial for $\bar{d}$ not too large. The function $K_c$ is then obtained by substituting $\beta_c$ in Eq. (48) or (49):

$$K_c(\bar{d}) = \frac{2}{(1+\bar{d})}\left[\cos\sigma(\bar{d}) - \sqrt{3}\sin\sigma(\bar{d})\right]^{-3} \approx \frac{27}{4\bar{d}^3\left(1+2\bar{d}+0.556\,\bar{d}^2\right)}, \quad (52)$$

where $\sigma$ is defined in Eq. (51b). The function $K_c$ is plotted in both Supplementary Fig. 3a, b. This expression of $K_c$ provides a simple necessary condition for the groove closure, i.e. $K < K_c$. Using the definition of $K$, this criterion can be written as

$$B < B_c = \frac{\gamma h\ell_c^2}{2}\,\bar{d}\,K_c(\bar{d}). \quad (53)$$

For a given liquid ($\gamma$) and a given depth ($h$) and width ($d$) of the groove, Eq. (53) gives the value of bending modulus of the flexible sheet below which neighbouring walls are brought into contact (the groove closes) when the liquid rises. Using the approximate expression of $K_c$ in Eq. (52), the criteria for the groove closure can be rewritten, up to order 2 in $d/h$, by the following inequality :

$$\Lambda \equiv \frac{8Bd^2}{27\gamma h^3\ell_c^2}\left[1 + 2\frac{d}{h} + 0.556\left(\frac{d}{h}\right)^2\right]^{-1} < 1. \quad (54)$$

Note that once rolled, the structure remains closed even during the second capillary rise. Indeed, a reopening would have an energetical cost as it would lead to an increase of the air-liquid interface. When the second imbibition takes place, the pulling contact line at the edges of the wall below the second capillary rise height disappears. However, these forces are still present above the second capillary rise height, and the depression within the liquid column maintains the bottom of the structure closed. \subsection{Short-time dynamics.}

Since Eq. (44a) is solved with the initial condition $\bar{\ell}_I = 0$ at $\bar{t} = 0$, Eq. (44b) implies that $\beta$ is vanishing at $t = 0$ and hence small at short time. Taking the limit $\beta \to 0$ in Eq. (44a) and in the right-hand side of Eq. (44b) and considering $\bar{\ell}_I \ll 1$ in the right-hand side of Eq. (44a), the system of Eqs. (44) becomes

$$2\,\bar{\ell}_I\,\dot{\bar{\ell}}_I = c(\bar{d}), \quad c(\bar{d}) = [\bar{d}^2 G(0,\bar{d})]^{-1}, \quad (55a)$$

$$\beta = \left[\frac{1+2\bar{d}}{K\bar{d}^2}\right]\bar{\ell}_I, \quad (55b)$$

where, for $\bar{d} \lesssim 1$,

$$G(0,\bar{d}) = \frac{24\sqrt{3}\,\bar{d}}{2\sqrt{3} - \bar{d}} \quad \Rightarrow \quad c(\bar{d}) = \frac{2\sqrt{3} - \bar{d}}{24\sqrt{3}\,\bar{d}^3}. \quad (56)$$

Eq. (55a) is easily integrated and yields

$$\bar{\ell}_I^{\rm st}(\bar{t}) = \sqrt{c}\,\bar{t}^{1/2}, \quad (57)$$

where the superscript 'st' stands for short-time. Substituting Eq. (57) in Eq. (55b), we get

$$\beta^{\rm st}(\bar{t}) = \left[\frac{\sqrt{c}\,(1+2\bar{d})}{K\bar{d}^2}\right]\bar{t}^{1/2}, \quad (58)$$

where $c$ is given by Eq. (56).

Supplementary Fig. 4a, b shows comparisons between these short-time behaviours of $\beta$ and $\bar{\ell}_I$ and the numerical solutions of Eq. (44), respectively. The dimensionless stiffness is set to $K = K_c(\bar{d})/2$ so that the groove closes and the system reaches the stationary solution $\beta = \bar{d}/2$ and $\bar{\ell}_I = \bar{\ell}_I^{\infty,\,\rm closed} \to 2 + \bar{d}$ at long time.

**Time needed for groove closure**

To determine the time needed for a groove to close, i.e. the time needed for $\beta$ to reach $\bar{d}/2$, Eq. (44) are first uncoupled to obtain an equation for $\beta$. To do so, $\bar{\ell}_I$ is obtained as a function of $\beta$ from Eq. (44b) and substituted in Eq. (44a):

$$2K^2 G_1(\beta,\bar{d})\,\beta\,\dot{\beta} = 1 + \beta - KG_2(\beta,\bar{d})\,\beta, \quad (59a)$$

$$G_1 = \bar{d}^2(\bar{d} - \beta)^3\,\frac{[\beta(\beta - 4\bar{d} - 2) + \bar{d}(1+2\bar{d})]}{(1+2\bar{d} - \beta)^3}\,G, \quad (59b)$$

$$G_2 = \frac{(\bar{d} - \beta)^2}{1 + 2\bar{d} - \beta}, \quad (59c)$$

where the function $G$ is given by Eqs. (28) and (29).

Since the function $G$ is strictly positive, the function $G_1$ vanishes only when the expression between square brackets in Eq. (59b) is equal to zero or, equivalently, when $\beta = \beta^*$ with

$$\beta^* = 1 + 2\bar{d} - \sqrt{1 + 3\bar{d} + 2\bar{d}^2} = \frac{\bar{d}}{2} + \frac{\bar{d}^2}{8} + \mathcal{O}(\bar{d}^3). \quad (60)$$

Therefore, as $\beta$ approaches $\beta^*$, which can never be reached because $\beta \le \bar{d}/2$, $G_1$ becomes small and $\dot{\beta}$ must become large to compensate it since the right-hand side of Eq. (59a) is finite (and strictly positive when $K < K_c$). Note that, because $\beta^* > \bar{d}/2$, $\beta$ can reach continuously its physical extreme value ($\bar{d}/2$ when $K < K_c$) without any discontinuous jump. Indeed, if $\beta^*$ was smaller than $\bar{d}/2$, $\beta$ would diverge at $\beta = \beta^*$ and $\beta$ would then jump from $\beta^* < \bar{d}/2$ to $\bar{d}/2$.

In summary, at short time, $\beta$ grows as $\bar{t}^{1/2}$, as shown by Eq. (58), and $\dot{\beta}$ decreases until, if $K < K_c$, $\beta$ approaches $\bar{d}/2$ where $\dot{\beta}$ sharply increases (and $\beta$ as well).

The closure time at which $\beta = \bar{d}/2$ can be obtained exactly. Indeed, Eq. (59a) is separable and can be integrated side by side yielding

$$2K^2\int_0^\beta \frac{G_1(\beta',\bar{d})\,\beta'}{1 + \beta' - KG_2(\beta',\bar{d})\,\beta'}\,d\beta' = t, \quad (61)$$

where the initial condition $\beta(0) = 0$ has been considered. Therefore, the closure time is simply obtained by replacing $\beta$ by $\bar{d}/2$ in this relation:

$$\bar{t}_{\rm closure} = 2K^2\int_0^{\bar{d}/2}\frac{G_1(\beta,\bar{d})\,\beta}{1 + \beta - KG_2(\beta,\bar{d})\,\beta}\,d\beta, \quad (62)$$

where $G_1$ and $G_2$ are defined by Eqs. (59b) and (59c) respectively. Unfortunately, we were unable to compute this integral exactly. Nevertheless, it can be approximated as follow.

As seen in Supplementary Fig. 4a, the intersection between the short-time behaviour of $\beta$ given by Eq. (58) and its stationary solution

$\beta = \bar{d}/2$ gives an estimation the time needed for a groove to close when $K < K_c$. Solving $\beta^{st}(\bar{t}) = \bar{d}/2$ and using the expression (56) of $c$, we have

$$\bar{t}_{closure}^{st} \sim \left[ \frac{6\sqrt{3}\,\bar{d}^9}{\left(2\sqrt{3}-\bar{d}\right)\left(2\bar{d}+1\right)^2} \right] K^2, \tag{63}$$

where the superscript 'st' stands for short-time, since this expression is obtained by extrapolating the short-time behaviour of $\beta$. However, Supplementary Fig. 4a shows that this expression overestimates the closure time obtained numerically. In addition, the closure time should diverge when $K$ tends to $K_c$ from below because the groove never closes when $K > K_c$. This divergence is not captured by Eq. (63). It is thus expected that Eq. (63) describes well the exact closure time (62) up to a factor and for $K \ll K_c$. However, Eq. (63) can be corrected to better describe the closure time obtained numerically:

$$\bar{t}_{closure}^{fit} = \frac{\bar{t}_{closure}^{st}}{2.4 - 0.2\ln(\bar{d})/\ln 2}\left[1 - \frac{K}{K_c}\right]^{-4/5}. \tag{64}$$

One factor, which varies slowly with $\bar{d}$, corrects the overestimation of the closure time mentioned above and the second (ad hoc) factor describes the divergence of the closure time near $K = K_c$. Supplementary Fig. 4d shows indeed that Eq. (63) overestimates the exact closure time when $K \ll K_c$ and strongly underestimates it when $K \simeq K_c$ since Eq. (63) does not diverge near this value. However, the corrected estimation (64) describes well the exact closure time for any $K$.

## Second capillary rise dynamics

When complete closure of the grooves occurs, a second capillary rise takes place in the core of the newly formed pipe. We assume the number of ribs to be large enough to consider that the pipe section is circular. The contact between neighbouring ribs sets the radius of the pipe, $R_{int} = wh/d$, but also increases the mechanical resistance of the structure[36] which can be considered as rigid.

The dynamics of capillary rise in such rigid cylindrical pipes has already been described by Lucas[6] and Washburn[7]. The flow is assumed unidirectional, and we neglect inertial forces that only play a role at very early dynamics. Then, the flow results from the competition between the driving capillary force, due to the pressure gradient induced by the curvature of the liquid-vapour interface, and the resistive gravitational and viscous forces. The flow in the pipe is Poiseuille-like, and the relation between the capillary pressure drop $\Delta p$ and the flow rate $Q = \pi R_{int}^2 d\ell/dt$ reads:

$$\Delta p - \rho g \ell_{II} = R_\mu Q, \tag{65}$$

where $R_\mu = 8\mu\ell_{II}/(\pi R_{int}^4)$ is the hydraulic resistance of a cylindrical pipe of radius $R_{int}$.

The pressure at the top of the liquid column is given by the Laplace law $p(\ell_{II}) = -2\gamma/R + p_{atm}$, while the pressure at the bottom of the pipe is equal to the atmospheric pressure $p(0) = p_{atm}$. Hence, the pressure drop is $\Delta p = 2\gamma/R$.

Using the expressions of the flow rate, $Q$, and of the capillary pressure drop, $\Delta p$, in Eq. (65), the evolution equation for liquid height $\ell_{II}$ in the tube:

$$\frac{8\mu}{R_{int}^2}\ell_{II}\dot{\ell}_{II} = \frac{2\gamma}{R_{int}} - \rho g \ell_{II}, \tag{66}$$

where $\dot{\ell}_{II} = d\ell_{II}/dt$. For convenience, the evolution Eq. (66) can be made dimensionless by rescaling the variables as follows :

$$\bar{\ell}_{II} = \ell_{II}/\ell_{II}^\infty, \quad \bar{t} = t/\tau_{II}, \tag{67}$$

where

$$\ell_{II}^\infty = 2\frac{\ell_c^2}{R_{int}}, \tag{68}$$

is the equilibrium height of the second capillary rise (i.e. the Jurin height) obtained by setting $\dot{\ell}_{II} = 0$ in Eq. (66), and $\tau_{II}$ is given by :

$$\tau_{II} = \frac{16\mu\gamma}{\rho^2 g^2 R_{int}^3}. \tag{69}$$

In dimensionless form, Eq. (66) becomes

$$\bar{\ell}_{II}\dot{\bar{\ell}}_{II} = 1 - \bar{\ell}_{II}. \tag{70}$$

This first-order nonlinear ODE can be solved numerically using MATLAB or Mathematica with the initial condition $\bar{\ell}_{II}(\bar{t}_{closure}) = 0$. For $\bar{t} < \bar{t}_{closure}$ no second capilary rise occurs as the whole structure has not deformed into a tube and the second capillary rise is set to zero, $\bar{\ell}_{II}(\bar{t} < \bar{t}_{closure}) = 0$.

## Full immersion of the structure into the liquid bath

In this section, we propose a model to describe the dynamics of the capillary rise in structures that are plunged in the liquid bath at a velocity $V$ until fully immersed and then withdrawn from this bath with the same velocity. We compare the cases of the immersion of a rigid structure, that has the same geometry as the tubular-shaped closed flexible structure, and of a flexible grooved sheet.

Let us first consider the case of a rigid structure. The situation differs from what has been described in the previous sections as, first, the structure being rigid, $\beta$ is constant and equal to $\beta = \bar{d}/2$, and second, while the structure is immersed into or withdrawn from the bath the pressure at the bottom of the structure varies with the length of immersion $L'$. In the immersion phase $L' = Vt$ with $0 \le t \le t^*$ and with $t^* = L/V$, the time at which the structure is fully immersed and withdrawing starts, while in the withdrawing phase $L' = V(2t^* - t)$, with $t^* \le t \le 2t^*$. Hence, the extra pressure drop is $\Delta p_{im} = \rho g V t$ during the immersion phase and $p_{im} = \rho g V(2t^* - t)$ in the withdrawing phase.

The dynamics of the capillary rise $\ell_I$ in the pipe of circular sector cross-section is thus described by :

$$\frac{\mu h}{2d^3}G(\beta=\bar{d}/2,\bar{d})\ell_I\dot{\ell}_I = \frac{2\gamma(2+\bar{d})}{d} + \rho g V t - \rho g \ell_I$$
$$\text{for } 0 \le t < t^*, \text{ and} \tag{71a}$$

$$\frac{\mu h}{2d^3}G(\beta=\bar{d}/2,\bar{d})\ell_I\dot{\ell}_I = \frac{2\gamma(2+\bar{d})}{d} + \rho g V(2t^* - t) - \rho g \ell_I$$
$$\text{for } t^* < t \le 2L/V. \tag{71b}$$

Note that these equations correspond to Eq. (30), for $\beta = \bar{d}/2$, and hence $A = hd/2$, to which is has been added the aforementioned extra pressure term. By rescaling $\ell_I$, $t$ and $V$ in Eq. (71) by $\ell_I^{\infty,open}$, $\tau_I$, and $\ell_I^{\infty,open}/\tau_I$, respectively, we obtain the dimensionless form of the evolution equation of the capillary rise into the closed grooves:

$$G(\beta=\bar{d}/2,\bar{d})\bar{d}^2\bar{\ell}_I\dot{\bar{\ell}}_I = (2+\bar{d}) + \bar{V}\bar{t} - \bar{\ell}_I$$
$$\text{for } 0 \le \bar{t} < \bar{t}^*, \text{ and} \tag{72a}$$

$$G(\beta=\bar{d}/2,\bar{d})\bar{d}^2\bar{\ell}_I\dot{\bar{\ell}}_I = (2+\bar{d}) + \bar{V}(2\bar{t}^* - \bar{t}) - \bar{\ell}_I$$
$$\text{for } \bar{t}^* < \bar{t} \le 2\bar{t}^*. \tag{72b}$$

These first-order nonlinear ODEs are solved numerically using MATLAB with the initial condition $\bar{\ell}_I(0) = 0$.

Following the same reasoning, Eq. (66) describing the capillary rise $\ell_{II}$ in the inner tube may be transformed to take into account the extra pressure term:

$$\frac{8\mu}{R_{\text{int}}^2}\ell_{II}\dot{\ell}_{II} = \frac{2\gamma}{R_{\text{int}}} + \rho g V t - \rho g \ell_{II} \tag{73a}$$
$$\text{for } 0 \le t < t^* \text{ and}$$

$$\frac{8\mu}{R_{\text{int}}^2}\ell_{II}\dot{\ell}_{II} = \frac{2\gamma}{R_{\text{int}}} + \rho g V(2t^* - t) - \rho g \ell_{II} \tag{73b}$$
$$\text{for } t^* < t \le 2L/V.$$

To obtain the dimensionless form of Eq. (73), we rescale $\ell_{II}$ by $\ell_{II}^{\infty}$, $t$ by $\tau_{II}$ and $V$ by $\ell_{II}^{\infty}./\tau_{II}$. It leads to

$$\bar{\ell}_{II}\dot{\bar{\ell}}_{II} = 1 + \bar{V}\bar{t} - \bar{\ell}_{II} \quad \text{for } 0 \le \bar{t} < \bar{t}^* \text{ and} \tag{74a}$$

$$\bar{\ell}_{II}\dot{\bar{\ell}}_{II} = 1 + \bar{V}(2\bar{t}^* - \bar{t}) - \bar{\ell}_{II} \quad \text{for } \bar{t}^* < \bar{t} \le 2\bar{t}^*. \tag{74b}$$

A MATLAB routine is used to solve these ODEs with the initial condition $\bar{\ell}_{II}(0) = 0$.

For the case of the immersion of the flexible grooved sheet, we consider that during the immersion phase $\ell_I = Vt$ since while the structure is plunged in the bath the structures remains open (see Supplementary Videos 5 and 6) and the liquid fills the immersed portion of the structure from the side of the grooves almost instantaneously. To model the capillary rise dynamics in the withdrawing phase, we assume that, at the transition between the immersion phase and the withdrawing phase (i.e. at $t = t^*$), the structure deforms into a tube, therefore, $\ell_{II}$ jumps to the value $\ell_{II}(t = t^*) = L$. Then, while the structure—that remains in its deformed state—is withdrawn from the bath (i.e. for $t > t^*$), the captured liquid drains. The equations that govern the drainage in the grooves and in the core of the newly formed tube are the same as in the rigid case, Eqs. (71b) and (73b), respectively. However, the initial conditions change and are $\ell_I(t = t^*) = \ell_{II}(t = t^*) = L$.

A MATLAB routine is used to solve the first-order ODEs and obtain the height of the first and second capillary rise at the end of the withdrawing phase, $\ell_I(t = 2t^*)$ and $\ell_{II}(t = 2t^*)$.

Examples of capillary rise dynamics during the entire dipping of flexible or rigid structures in the liquid bath are shown in Fig. 3c of the article and in Supplementary Videos 5 and 6, and the evolution of the final capillary rise heights as a function of the velocity $V$ is illustrated in Fig. 3d of the article.

## Data availability

The data supporting the findings of this study and the material are provided in the Source Data file and are available on request from the corresponding authors. Source data are provided with this paper.

## Code availability

A supporting text describing the modelling procedures is available in the 'Methods'. The code used for numerical integration is provided with the paper and available from the corresponding authors on request. MATLAB scripts used to numerically solve the fluid-structure interaction are available on the following link: https://doi.org/10.5281/zenodo.15411342.

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

## Acknowledgements

We acknowledge support by F.R.S.-FNRS under the research grants CR n°40017301 'G.El.In.Flow' (J.C.) and CR n°40011004 'BioCapTure' (E.S.). This project has received funding from the European Union's Horizon 2020 research and innovation programme under the Marie Sklodowska-Curie grant agreement no. 101027862 (E.S.) and from 747 Fédération Wallonie-Bruxelles (F.B., ARC ESCAPE project). We thank the Micro-milli service platform (ULB) for access to their experimental facilities.

## Author contributions

E.S. and J.C. designed research; E.S. and J.C. performed research; E.S. and J.C. analysed data; E.S., J.C. and F.B. developed the model; E.S., J.C., F.B. and B. S. wrote the manuscript.

## Competing interests

E.S., F. B., B. S. and J. C. are co-authors on a filed patent No. 24199680.0, which describes the methods used herein.

## Ethics approval

All people who contributed substantially to this manuscript were invited to join as co-authors. No ethics approval was required for this research.
