## [Transparent Peer Review file · Nature Communications]

Elastocapillary sequential fluid capture in hummingbird-inspired grooved sheets

Corresponding Author: Dr Emmanuel SIEFERT

Version 0:

Reviewer comments:

Reviewer #1

(Remarks to the Author)

Key Results

The study introduces a flexible fluidic device inspired by a hummingbird's tongue, aimed at fast liquid transfer, measuring blood hematocrit levels and some other medical applications. The device is presented as a potential competitor to traditional measurement tools like centrifuges and paper-based microfluidics. However, a detailed comparison of the device's speed and accuracy with established methods is lacking. The authors also explore the device's mechanical properties, such as its ability to reopen after compression, which could have practical implications for reusable, flexible, fluidic technologies. However, some theoretical and experimental clarifications are required to assess the device's performance fully.

Validity

While the study presents promising results, there are notable data interpretation and validation gaps.

The authors claim the device's superiority over existing methods but do not provide sufficient quantitative comparisons on accuracy or speed. Quantitative comparisons with existing devices are needed. In Figure 4c and Video 9, the authors mention a direct correlation between blood viscosity and hematocrit levels (also noted in reference 44). However, since their device is not a viscometer, how do they propose to calibrate hematocrit levels against viscosity to quantify the accuracy and speed of their measurements? Please clarify.

Additionally, On page 9, the authors assume a no-slip boundary condition. Is this assumption always valid for a silicone oil-infused elastomer surface?

In Figure 2c, I am unable to distinguish I_I from I_{II}. Also, for $e=230\ \mu\text{m}$, why does the theoretical line deviate from the experimental data?

Significance

The potential impact of this flexible fluidic device could be significant for applications requiring portable and reusable blood measurement tools. However, the lack of comprehensive comparison data, such as speed and accuracy relative to existing devices, limits the study's immediate influence on the field. Addressing these gaps (in the form of table or figure) would considerably enhance its significance, particularly in medical diagnostics and field testing scenarios.

On page 14, the authors state that their flexible device "may be reopened when compressed." Instead of using "may be," can evidence be provided? If this is beyond the study's scope, the statement should be removed; otherwise, please support it with evidence.

Data and Methodology

The methodology used in the study appears sound but needs more clarification. In Extended Data Figure 1, the figure labels it as an LED panel, but the caption refers to an LCD backlight. Please clarify which one is correct. Additionally, please specify the purity of the silicone oil used. Please include a length scale in Figure 2b; it is not mentioned in the caption.

Theoretical Approach

In Equations 4 and 5 of the main text, two systems of coupled nonlinear ODEs are presented, yet only one initial condition is specified. Shouldn't there be an additional initial or boundary condition? Please clarify. On page 6, three different scenarios are presented. Although the authors provide a theoretical expression for the increase in captured volume at the end of page 7, could they also offer experimental quantification or relate existing figures to this theoretical statement?

Clarity and Context

The manuscript is generally clear, but some areas could benefit from additional explanation and context. It's unclear whether silicone oil must always be used as the wetting liquid to achieve a Young's contact angle of 0° . For instance, if blood samples are to be tested, would the wetting liquid remain silicone oil, or would blood serve as the wetting medium? It's difficult to distinguish between retraction speed and volume since both are denoted as "V." In the Experimental Section on page 21, why were two different frame rates (100 fps and 120 fps) used?

Suggested Improvements

Please address all the points mentioned above in the manuscript.

Reviewer #2

(Remarks to the Author)

The authors present a passive microfluidic pipetting system made of a grooved elastic sheet inspired from hummingbird's tongue. The system can be used either by dipping only its lower end (scarce liquid) or by immersing it fully (abundant liquid). In the former case, the captured volume can be increased. The capillary imbibition taking place at the level of the individual grooves can - under certain conditions - trigger the rolling up of the sheet on itself. By adjusting the sheet width, a close tube can be formed, and a second capillary imbibition process can develop at this larger scale. Compared to a rigid (flat) system of the same geometry, a larger volume of liquid can be captured. For abundant liquid, the system, which is first immersed in a liquid bath, offers a faster capture. The imbibition of the small grooves is not limited by viscosity as the rolled-up structure appears only when pulled in air. In contrast, for a rigid (rolled-up) system of the same geometry, the imbibition requires the liquid to flow along the main groove direction, reducing the capture speed. Thus, the system presents for both scarce and abundant liquid configurations, important advantages compared to its rigid flat and rolled counterparts.

The concept, experimental results, and modelling are interesting and convincing and are worth being published. Yet, the paper in its current form suffers from some weaknesses that must be addressed prior to publication.

One important issue is the paper organization. While concision in the presentation of the main results is appreciable, it does not deserve the work best here as (i) the paper is difficult to follow, (ii) important results are only presented in the extended figures and (iii) claims about possible applications are not appropriately justified.

Other points are raised below in the order they appear.

Essential elements such as the geometry of the sheets with variables e , d , h , w , L are spread over different figures with many sub-figures and small insets (especially in fig 1a) and are also not always clearly described in the text. I would recommend adding a clear and big enough figure to introduce all these parameters at once at the beginning. This may require a reorganization of the figures (see also my comments about figure 4 and extended figure 3 at the end). It would be good to have a table with the typical values/tested ranges. It is not clear how much e and w have been varied and how they compare to d and h . Similarly, it seems that at least two viscosities and surface tensions have been used (two silicon oils and a water solution). The properties of the liquid used for obtaining the presented data should be listed in a table.

Figure 3 is referred to before figure 2, this should be changed.

Figure 2 d: the log scale compresses the region of interest where the transition occurs so we cannot really see how good the agreement is. Could the scale be changed or a zoom added in an inset for example?

For the torque M_γ , the "top of the wall" (page 6 line 92) is not immediately clear. You may reformulate or add a vectorial expression first with the lever and force. Same for the other torque.

About the closure of the sheet and the conditions derived from the balance between the capillary torque/elastic bending (page 6): if I understood correctly, the criterion is established considering walls with a wet side (on the wet groove side) and a dry side (outside on the edge) - meaning that the system closes from the edge of the sheet. Is that correct? If not, can you explain why for a wall between two adjacent and equally filled grooves, the torques on each side do not compensate. What maintains the system close once it is rolled (so when there is no groove on the edge)? What maintains the system close once the second imbibition takes place (so when the contact line at the "top of the wall" disappears)? Could the authors add a discussion on these points?

The bending stiffness of an elastic sheet varies with its thickness at power 3. Here (see B, page 6, line 95) the authors chose to take this thickness equal to e . Can they justify why the thicker sections (width w , thickness $e+h$) can be neglected and if a second order effect be expected?

End of page 7, could the authors add typical values of the volume captured with the flat/rolled geometries of a "reference geometry". What's the relative gain?

Page 8, line 123, t_{closure} is discussed without having been introduced.

Page 12, line 186 "In conclusion" is confusing here since figure 4 has not yet been discussed or even mentioned at this stage.

Figure 4: This figure provides very little information compared to figures 1, 2 and 3 as well as the extended figure 3 (especially 3c). It is mostly here to illustrate claims about possible applications, which should however either been better justified or reduced (see below). I would suggest keeping data of sub-figure c (graphs only) and add them to another figure. I would also strongly consider adding extended figure 3 or at least the subfigure 3c to the main text as the closing of the sheet is at the core of this work.

Finally, as mentioned, I have some comments about biomedical applications. Most importantly, it would be needed to specify for which kind of point-of-care applications it could be used. For blood analysis with immunoassays, reactions are

carried out on the plasma only so that the centrifugation applied in the case of lab-on-a-disk devices for examples is needed and not to be suppressed. Those disks are very cheap and offer the possibility to have sequential reactions with stored reactants, valves, mixing and extraction processes... If compared to small glass/plastic capillaries used to sample blood as illustrated in Figure 4a, I am not sure that the storage argument is so strong (page 14) and the flexibility of the sheet could induce difficulties in manipulation compared to a rigid capillary. It is probably more appropriate to compare the proposed system to paper microfluidic. Here again, I am a bit missing arguments for the moment – especially if as I have no hydrophilic and environment friendly elastomers in mind. Could the authors better explain the advantages of their systems over established ones or lower their claims?

Reviewer #3

(Remarks to the Author)

See comments.

Version 1:

Reviewer comments:

Reviewer #1

(Remarks to the Author)

No comments

(Remarks on code availability)

Reviewer #2

(Remarks to the Author)

I had two major comments related to (i) the organization and readability of the article and (ii) the strength of the claims regarding potential applications for Point of Care (POC).

Regarding the first point (i), the authors have carefully addressed the questions and reorganized the manuscript and figures, which has significantly improved the paper.

I still have a comment about figure readability — I_l (the letter, not the symbol when plotted in a graph) is barely visible on my screen. Perhaps using bold fonts or a slightly darker blue would improve visibility.

Furthermore, the authors have significantly toned down their claims, making them more aligned with the current state of their research. However, presenting the need to make a hydrophobic elastomer hydrophilic through a subsequent surface treatment — whose effectiveness is limited to just one day — as a possible solution rather than a potential challenge remains misleading. In real applications, where production costs, shelf life, and reproducibility are crucial, this remains a critical issue and I would appreciate if a remark could be added to underline this limitation.

I have no further comments beyond these two points, which could quickly be addressed prior to publication.

(Remarks on code availability)

Reviewer #3

(Remarks to the Author)

I appreciate the responses to my inquiries.

(Remarks on code availability)

Point-by-point Response to the Reviewers

We thank the reviewers for their time and their feedback on our manuscript. Please find below our point-by-point responses to the reviewers' comments.

Response to Reviewer 1:

Reviewer 1 states: “The study introduces a flexible fluidic device inspired by a hummingbird’s tongue, aimed at fast liquid transfer, measuring blood hematocrit levels and some other medical applications. The device is presented as a potential competitor to traditional measurement tools like centrifuges and paper-based microfluidics. However, a detailed comparison of the device’s speed and accuracy with established methods is lacking. The authors also explore the device’s mechanical properties, such as its ability to reopen after compression, which could have practical implications for reusable, flexible, fluidic technologies. However, some theoretical and experimental clarifications are required to assess the device’s performance fully.”

Our Answer: We thank the reviewer for their thorough review of our paper. We added some critical experiments to better assess the application potential of our device and have built on the specific comments and suggestions of the reviewer to improve the clarity of our manuscript and avoid any misunderstanding.

We would like to highlight the fact that the key result in this study is not the application but rather the new physical phenomenon at play: it consists in the spontaneous curling of a textured sheet when plunged (deeply or shallowly) into a liquid bath due to elastocapillary forces, that enables fast and augmented liquid capture. We accurately model and rationalise the fluid-structure interaction dynamics and quantitatively compare the performance of the flexible device to rigid counterparts. As an opening to the study, we give some proofs of concept of potential use of the studied structure. Yet, direct comparison with existing measurement tools is beyond the scope of this study and is the focus of a current effort in the team, following our patent application, that would lead to a way more technical and application-oriented study.

Please find below our detailed answer to the referee’s comments.

Reviewer 1 states: “While the study presents promising results, there are notable data interpretation and validation gaps. The authors claim the device’s superiority over existing methods, but do not provide sufficient quantitative comparisons on accuracy or speed. Quantitative comparisons with existing devices are needed.”

Our Answer: We agree with reviewer 1 that superiority claims are, at this stage, exag-

gerated and we thank them for pointing this out. Our writing was indeed too enthusiastic in the conclusion, and one sentence was problematic, stating that our approach “could compete with centrifugal microfluidics”.

In the corrected version of the article we revised our claims and we also better demonstrate the key benefits of the approach: (i) a passive change in channel shape that allows for fast initial liquid transport and a protected liquid sample in the deformed state, (ii) an augmented volume captured thanks to deformability.

Changes to the manuscript. We changed the conclusion sentence, removing any comparison and superiority claim with respect to existing approaches: “As a whole, our device has the potential to become an all-in-one fluidic testing platform and may offer an alternative approach to existing testing techniques, such as centrifugal microfluidics for at least blood testing, as it is passive, fast, affordable, and requires a small volume sample. These attributes are also compatible with those of paper-based microfluidics making medical diagnostics accessible for development countries.” (lines 296-301).

To go one step further, we added an experimental evidence that our device can be effectively used to capture blood (Figure 5a). This is enabled by carrying out a surface treatment of the initially hydrophobic silicone membrane : first a plasma treatment is achieved, making the surface hydrophilic on short times, then, the grooves are filled with a solution of serum albumin at 2wt%, and last, the serum albumin solution is dried by putting the imbibed structure in an oven at 50°C during one hour. After treatment the surface remains hydrophilic for at least a couple of days and can be used for the capture of any aqueous liquid. We also added a sentence stating that, the perspective of using the device for ABO test would require further development to fix antibodies in the grooves (lines 289-292): “We believe that replacing the dye with dehydrated antibodies in the grooves, which requires additional work beyond the scope of this study, should allow rapid blood tests using only a fraction of a drop volume instead of the three droplets needed, for example, in a Serafol® ABO+D card used to confirm patient identity immediately before a blood transfusion.” Yet, making the device hydrophilic open the perspective of depositing a solution of antibodies into the structure’s grooves. After drying out the solution, the prepared structure could be used for blood testing.

We also completed the quantitative comparison between our flexible device and rigid counterparts, which was done so far only in Figure 4 (Fig. 3 in the previous version of the manuscript).

- We added a Figure panel (Fig. 2e), in which we quantitatively compare the amount of liquid captured depending on the deformability of the device.

- We added a Figure panel (Fig. 3d), in which we show that, for the specific groove aspect ratio considered in this study, the hydraulic resistance increases by more than a factor 6 between the open and closed (deformed) state, allowing fast liquid transport in the early stages of fluid capture and a protected liquid sample after deformation.

Reviewer 1 states: “In Figure 4c and Video 9, the authors mention a direct correlation between blood viscosity and hematocrit levels (also noted in reference 44). However, since their device is not a viscometer, how do they propose to calibrate hematocrit levels against viscosity to quantify the accuracy and speed of their measurements? Please clarify.”

Our Answer: Reference [44] in the main manuscript, along with Ref.^{1,2} (see below), establish a direct relationship between hematocrit levels and blood viscosity. Specifically, References^{1,2} provide correlation curves that enable the determination of hematocrit levels based on blood viscosity measurements. Given that the closure time of our system is directly proportional to fluid viscosity, the device effectively functions as a viscometer. Note that Ostwald viscometers or falling-sphere viscometers work on the same principle, measuring a time or a speed that is, for one given device, directly linked to a viscosity. To determine viscosity, equation (64) of the manuscript can be utilized, provided the Young’s modulus of the rubber, the geometric dimensions of the structure, and the surface tension of the liquid are known. Alternatively, a simpler approach would involve calibrating the device’s closure time using blood samples with known hematocrit levels. The resulting correlation curves could then be used to determine the hematocrit level of the blood under test.

Changes to the manuscript. To avoid any misunderstanding, we moved the demonstration of potential use of the device as a viscosity probe in Fig. 3. We thus separate the physical explanation, in which we show that the closure time increases linearly with the fluid viscosity (lines 210-212), and the potential application (using this property) to measure the viscosity of biological fluids (e.g. blood), giving access to information about the fluid content (e.g., hematocrit levels) (lines 294-296).

Reviewer 1 states: Additionally, On page 9, the authors assume a no-slip boundary condition. Is this assumption always valid for a silicone oil-infused elastomer surface?

Our Answer: The referee raises an excellent question. For the sake of simplicity we consider that there is no slip length in our problem. The good agreement between experiments and theory supports the fact that the assumption is reasonable and that a refined and more complex model of the fluid-surface contact is not necessary to describe the double capillary rise dynamics. If a non-zero slip length were present, it would likely correspond to the size of the polymer mesh or

the polysiloxane chains of the silicone oil. Both are of the order of few tens of nanometers, which is negligible compared to the other characteristic lengths involved in the system. Therefore, we have disregarded the slip length in our analysis.

Changes to the manuscript. We added a short discussion concerning the choice of a non-slip boundary condition in the Methods, section ‘Flow in a groove’ : “We assume a usual non-slip boundary condition at the liquid-solid interface. This assumption is justified even if the solid has been swollen using the same liquid (silicone oil V10) in most of the experiments. Indeed, if a non-zero slip length is present, it would likely correspond to the size of the polymer mesh or the polysiloxane chains of the silicone oil. Both are of the order of few tens of nanometers, which is negligible compared to the other characteristic lengths involved in the system.”.

Reviewer 1 states: In Figure 2c, I am unable to distinguish ℓ_I from ℓ_{II} . Also, for $e = 230\mu\text{m}$, why does the theoretical line deviate from the experimental data?

Our Answer: We thank the reviewer for pointing out the lack of readability of the Figure. Concerning the relative disagreement between the experiment curves and the model observed in Fig. 2c $e = 230\mu\text{m}$ (Fig. 3a of the revised manuscript), we impute this to the fact that there are small variations of the groove dimensions across the membrane leading to differences in capillary rise along the various grooves, that are enhanced by the finite deformability of the separating walls: a slightly smaller groove will generate a larger depression in the rising liquid column, deflecting the wall towards the groove and closing this groove more than the neighbouring slightly larger groove. Close to the transition, such effects are amplified.

Changes to the manuscript. We changed the colours of the curves to increase the readability. We also added a short discussion in the manuscript to discuss the variability in capillary rise not captured by the model: “Note that near the closing transition ($1 \leq \Lambda \leq 8$), there is significant variability in the final capillary rise, ℓ_I^∞ . This variability, which is also visible in the second image of Fig. 2a, occurs within the same sheet (i.e., for the same Λ) and is not accounted for by our model, which assumes identical behavior for each groove. We attribute this variability to small differences in groove sizes, which can substantially affect the equilibrium height. Additionally, these size differences can result in uneven forces acting on either side of the wall, potentially causing bending and symmetry-breaking events. Such effects, commonly observed in elastocapillary phenomena, greatly amplify variations in the groove cross-sections and, consequently, the equilibrium capillary rise height.” (lines 158-166).

Significance

Reviewer 1 states: “The potential impact of this flexible fluidic device could be significant for applications requiring portable and reusable blood measurement tools.”

“However, the lack of comprehensive comparison data, such as speed and accuracy relative to existing devices, limits the study’s immediate influence on the field. Addressing these gaps (in the form of table or figure) would considerably enhance its significance, particularly in medical diagnostics and field testing scenarios.”

Our Answer: This study investigates a novel structure that exhibits an intrinsically intriguing shape-shifting behaviour driven by elastocapillary coupling. Following a detailed presentation of our experimental observations and the development of an analytical model, we propose potential applications for the device. Considering that the device could address several challenges in point-of-care testing (POCT), we provide proof-of-concepts to demonstrate this potential. However, an extensive comparison between the proposed device and existing technologies is beyond the scope of this paper.

Changes to the manuscript. We completely reformulate the discussion about potential applications, delete the sentences that could have been misleading, and clarify that the system could –after evident further development – offer alternatives to the more installed techniques. (lines 289-301).

Reviewer 1 states: “On page 14, the authors state that their flexible device “may be reopened when compressed.” Instead of using “may be,” can evidence be provided? If this is beyond the scope of the study, the statement should be removed; otherwise, please support it with evidence.”

Our Answer: We agree with the referee that the affirmation was not supported by any experimental proof. We now prove the assertion.

Changes to the manuscript. We have added Supplementary Video 7 in which we show that the structure can be easily manually reopened. A reference to the Supplementary Video 7 has been added in the article (lines 267-268).

Data and Methodology

Reviewer 1 states: “The methodology used in the study appears sound but needs more clarification.”

“In Extended Data Figure 1, the figure labels it as an LED panel, but the caption refers to an LCD backlight. Please clarify which one is correct. Additionally, please specify the purity of the silicone oil used.”

Our Answer: We thank the referee for spotting the error. A LED panel was used throughout the study. We have corrected it. We used standard silicone oils from Merck (Sigma Aldrich) for which data on purity are not available.

Reviewer 1 states: “Please include a length scale in Figure 2b; it is not mentioned in the caption.”

Our Answer: We thank the referee for spotting the error. We have added it on the new Fig. 2a.

Theoretical Approach

Reviewer 1 states: “In Equations 4 and 5 of the main text, two systems of coupled nonlinear ODEs are presented, yet only one initial condition is specified. Shouldn’t there be an additional initial or boundary condition? Please clarify.”

Our Answer: Equations 4 and 5 are a system of one ODE coupled with one algebraic equation. Hence, only one initial condition is needed. The initial value of β may be derived implicitly from Eq. (5) with the initial condition $\ell_I(t = 0) = 0$, yielding $\beta(0) = 0$.

Changes to the manuscript. A clarification has been added in the text lines 182-183: “Note that only one initial condition is needed since the second equation is algebraic.”

Reviewer 1 states: “On page 6, three different scenarios are presented. Although the authors provide a theoretical expression for the increase in captured volume at the end of page 7, could they also offer experimental quantification or relate existing figures to this theoretical statement?”

Our Answer: We thank the reviewer for raising this point. We now show the agreement between experiments and theory regarding the volume captured in Fig. 2e. The captured volume was determined through image analysis by measuring both the mean closing angle, β , derived from the apparent width of the structure, Δ (see Fig. 2a), and the average equilibrium rise height.

Changes to the manuscript. We added a panel in Fig. 2e that compares the volume captured by five structures of varying membrane thickness with the expected captured volume derived from our model. Moreover, a description regarding measurement of the captured volume from the experimental pictures is now given in Methods “Derivation of the captured volume from image analysis”.

Clarity and Context

Reviewer 1 states: “The manuscript is generally clear, but some areas could benefit from additional explanation and context.”

Our Answer: We thank the reviewer for the remark. By taking into account the suggestions of all three reviewers, we believe that the manuscript has now strongly improved in terms of clarity and organisation.

Reviewer 1 states: “It’s unclear whether silicone oil must always be used as the wetting liquid to achieve a Young’s contact angle of 0° .”

Our Answer: A Young contact angle of 0° reflects the liquid’s strong affinity for the surface. In the new Fig. 5, we demonstrate that water can act as a wetting liquid following a surface treatment (such as plasma treatment) applied to the silicone membrane to render it hydrophilic. Alternative treatments, such as coating with serum albumin, can also be used to achieve surface hydrophilicity and create a nearly zero contact angle between an aqueous solution and the membrane. However, it is important to note that a 0° contact angle is not essential to observe the described phenomenon. The critical requirement is that the liquid wets the surface (i.e., has a contact angle smaller than 90°) to observe the first capillary rise, which, if the structure is sufficiently flexible, triggers the structure closure and the subsequent second capillary rise. In supplementary Videos 7, we show that the same phenomenon can be obtained with ethanol (that has a contact angle of approximately 30° on our silicon rubber).

Changes to the manuscript. We added the new Fig. 5, in which panel a, proves that the system can be used with other liquid than silicone oil. Indeed, in Fig. 5a, the system is efficiently used to capture blood. This is enabled by achieving a surface treatment of the device. Details on the surface treatment are given in Method section ‘Surface treatment of the elastomer’.

Reviewer 1 states: “For instance, if blood samples are to be tested, would the wetting liquid remain silicone oil, or would blood serve as the wetting medium?”

Our Answer: Related to the above answer, if blood sample is tested, blood serves as the

wetting liquid. However, a necessary surface treatment of the silicone membrane must first be performed to make the surface hydrophilic.

Changes to the manuscript. The new Fig. 5a, shows the situation where blood is captured after surface treatment with albumin.

Reviewer 1 states: “It’s difficult to distinguish between retraction speed and volume since both are denoted as “V.””

Our Answer: We thank the reviewer for the remark and changed the notation of the volume to \mathcal{V} .

Reviewer 1 states: “In the Experimental Section on page 21, why were two different frame rates (100 fps and 120 fps) used?”

Our Answer: There is no reason for using two different frame rates. Both frame rates are high enough to observe the dynamics of the capillary rise, it simply depends on the camera setting used that day.

Suggested Improvements Reviewer 1 states: “Please address all the points mentioned above in the manuscript.”

Our Answer: We did our best to address all the points highlighted by the reviewer and hope that the reviewer is convinced that the paper now merits publication.

Response to reviewer 2.

Reviewer 2 states: The authors present a passive microfluidic pipetting system made of a grooved elastic sheet inspired from hummingbird’s tongue. The system can be used either by dipping only its lower end (scarce liquid) or by immersing it fully (abundant liquid). In the former case, the captured volume can be increased. The capillary imbibition taking place at the level of the individual grooves can - under certain conditions – trigger the rolling up of the sheet on itself. By adjusting the sheet width, a close tube can be formed, and a second capillary imbibition process can develop at this larger scale. Compared to a rigid (flat) system of the same geometry, a larger volume of liquid can be captured. For abundant liquid, the system, which is first immersed in a liquid bath, offers a faster capture. The imbibition of the small grooves is not limited by viscosity

as the rolled-up structure appears only when pulled in air. In contrast, for a rigid (rolled-up) system of the same geometry, the imbibition requires the liquid to flow along the main groove direction, reducing the capture speed. Thus, the system presents for both scarce and abundant liquid configurations, important advantages compared to its rigid flat and rolled counterparts.

The concept, experimental results, and modelling are interesting and convincing and are worth being published. Yet, the paper in its current form suffers from some weaknesses that must be addressed prior to publication.

One important issue is the paper organization. While concision in the presentation of the main results is appreciable, it does not deserve the work best here as (i) the paper is difficult to follow, (ii) important results are only presented in the extended figures and (iii) claims about possible applications are not appropriately justified. Other points are raised below in the order they appear.

Our Answer: We thank the reviewer for their positive feedback on the content of the article and on the constructive remarks for improving the manuscript organization and clarity. We reorganised the paper to make it easier to read, we replaced contents (initially in the "Methods" section) in the main text, and we added some supplementary experiments to better assess the application potential of our device. Please find below our detailed answer to the referee's comments.

Reviewer 2 states: Essential elements such as the geometry of the sheets with variables e , d , h , w , L are spread over different figures with many sub-figures and small insets (especially in fig 1a) and are also not always clearly described in the text. I would recommend adding a clear and big enough figure to introduce all these parameters at once at the beginning. This may require a reorganization of the figures (see also my comments about figure 4 and extended figure 3 at the end).

Our Answer: We thank the reviewer for their suggestion. We completely reorganized the Figures to follow the remarks.

Changes to the manuscript. The geometry of the sheet is now clearly presented in Fig. 1.

Reviewer 2 states: It would be good to have a table with the typical values/tested ranges. It is not clear how much e and w have been varied and how they compare to d and h . Similarly, it seems that at least two viscosities and surface tensions have been used (two silicon oils and a water solution). The properties of the liquid used for obtaining the presented data should be listed in a table.

Our Answer: We thank the reviewer for their relevant remark. In this system, there are six lengths (h , e , w , d , L and the capillary length $\ell_c = \sqrt{\gamma/[\rho g]}$). Varying independently all five dimensionless geometrical parameters would have been prohibitive. Instead, we made the decision to set the groove geometry (h , w , L and d) and to systematically vary the membrane thickness e , since it is the key geometrical parameter involved in the elastic deformation of the groove. We made this decision since the flow within open rigid groove has been thoroughly studied in recent years (see Refs. ³⁻⁵) and no surprises are expected to come from this side.

We additionally made a few experiments with other groove geometry (see Fig. 5a right panel, and Supplementary video S1, S5 and S7) and show that the same phenomenon is observed, but did not conduct a detailed analysis on these objects.

Changes to the manuscript: We added a table in the methods section ‘Properties of the fluid structure couples used in the article’, in which we list the range of the different parameters tested in this study.

Reviewer 2 states: Figure 3 is referred to before figure 2, this should be changed.

Our Answer: We thank the reviewer for spotting the error. We rearrange the Figures to correct this.

Changes to the manuscript. We changed Fig. 1 to highlight the different situations of use of the device and ease the references to these situations in the introduction.

Reviewer 2 states: Figure 2 d: the log scale compresses the region of interest where the transition occurs so we cannot really see how good the agreement is. Could the scale be changed or a zoom added in an inset for example?

Our Answer: Contrary to the referee’s statement, a logarithmic scale expands the region near $\Lambda = 1$ compared to a linear scale. For comparison, we present Fig. 2d in a linear scale just below. Therefore, we retain the original version of the plot.

Close to the transition, there is a large variability in one single sheet between neighboring groove, that the model cannot take into account, since the interaction between grooves is overlooked. We interpret this large variability near the critical point from the fact that small geometrical differences between neighboring grooves associated with the compliance of the separating walls leads to a symmetry breaking event, such that some grooves completely close whereas other do not. A few sentences have been added in the main text to better explain this shortcoming of the model (lines 158-166 and 186-187).

Reviewer 2 states: For the torque \$M_\gamma\$, the “top of the wall” (page 6 line 92) is not immedi-

Figure 1: Final equilibrium elevations ℓ_I and ℓ_{II} as a function of the dimensionless transition parameter Λ defined by Eq. (5) of the article in a linear scale.

ately clear. You may reformulate or add a vectorial expression first with the lever and force. Same for the other torque.

Our Answer: We thank the Reviewer for their remark.

Changes to the manuscript. We clarified the torque balance, using vectorial expression and a more precise vocabulary (lines 114-130).

Reviewer 2 states: About the closure of the sheet and the conditions derived from the balance between the capillary torque/elastic bending (page 6): if I understood correctly, the criterion is established considering walls with a wet side (on the wet groove side) and a dry side (outside on the edge) – meaning that the system closes from the edge of the sheet. Is that correct? If not, can you explain why for a wall between two adjacent and equally filled grooves, the torques on each side do not compensate.

Our Answer: In the situation studied in the article, all the grooves are filled: there is no "dry side" of the walls. In our model, we consider an elementary system: two walls linked together by a thin sheet. The deformable element of the system is the sheet of thickness e and width d and the walls are considered as rigid. When capillary rise occurs, a depression takes place within the liquid. This depression results in a force pulling on the walls, that generates a torque on the sheet's edges. Moreover, the force at the contact line at the top of the wall results in a second torque at the sheet's edge. The total torque is thus applied at the sheet's edge (see the red curled arrows in Fig. 2b), leading to its bending. The pressure and contact line forces of course are applied on the wall on each side, and thus, as the Reviewer rightfully points out, compensate: the wall does not deflect. So, these forces only translate into a torque acting on the edge of each sheet.

Another way to think about this system, that is equivalent and maybe more intuitive than

considering forces, is to consider the energy of the system. At equilibrium, we have two competing energies: (i) an interfacial energy, linked to the surface tension of the liquid, that promotes the closing of the groove to reduce the air-liquid interface area and (ii) an elastic energy, that resists the closing of the groove. When the bottom sheet is sufficiently soft (or thin), the groove will close, no matter whether the neighbouring groove is filled with liquid or not.

Changes to the manuscript. We added a sentence to clarify this point: “Note that the forces \mathbf{f}_γ and \mathbf{f}_p acts on each side of the walls and thus compensates, leading to no deflection of the walls.” (lines 130-131).

Reviewer 2 states: What maintains the system close once it is rolled (so when there is no groove on the edge)? What maintains the system close once the second imbibition takes place (so when the contact line at the “top of the wall” disappears)? Could the authors add a discussion on these points?

Our Answer: Related to our previous answer, we would like to emphasize the fact that the closing of the structure is not governed by the edges of the structure: each groove closes simultaneously due to the torque balance on the bottom sheet given by Eq. (3) in the manuscript. Once rolled, the reopening of the structure would have an energetical cost as it would lead to an increase of the air-liquid interface. Hence, to reduce the free energy, the structure remains closed. When the second imbibition takes place, there is indeed no pulling contact line at the edges of the wall below the second capillary rise height. However, these forces are still present above the second capillary rise height, and the depression within the liquid column maintains the bottom of the structure closed.

Changes to the manuscript. We added a short discussion at the end of Methods section ‘Stationary solution and condition for groove closure’.

Reviewer 2 states: The bending stiffness of an elastic sheet varies with its thickness at power 3. Here (see B, page 6, line 95) the authors chose to take this thickness equal to e . Can they justify why the thicker sections (width w , thickness $e+h$) can be neglected and if a second order effect be expected?

Our Answer: Related to the previous answers, the thickness e is taken in the model because it is the bottom sheet that experiences the capillary torque discussed above. The walls between the grooves are subjected to capillary forces that compensate, and thus do not deform.

Reviewer 2 states: End of page 7, could the authors add typical values of the volume captured with the flat/rolled geometries of a “reference geometry”. What’s the relative gain?

Our Answer: We thank the Reviewer for pointing out this issue.

Changes to the manuscript. We added a panel in Fig. 2e that shows the variation of the captured volume of liquid as a function of the membrane flexibility. The asymptotic plateau at high Λ (*i.e.* large e) corresponds to the rigid flat case. From the curve, one can see that for the device geometry used in Fig. 2a, the relative gain of captured volume is about 1.7.

Reviewer 2 states: Page 8, line 123, t_{closure} is discussed without having being introduced.

Our Answer: We thank the Reviewer for bringing this point to our attention.

Changes to the manuscript. t_{closure} is now defined lines 196-197 as the “time the structure takes to fully close after having being put in contact with the liquid bath”.

Reviewer 2 states: Page 12, line 186 “In conclusion” is confusing here since figure 4 has not yet been discussed or even mentioned at this stage.

Changes to the manuscript. We removed the problematic wording, and added sections in the article to bring clarity.

Reviewer 2 states: Figure 4: This figure provides very little information compared to figures 1, 2 and 3 as well as the extended figure 3 (especially 3c). It is mostly here to illustrate claims about possible applications, which should however either been better justified or reduced (see below). I would suggest keeping data of sub-figure c (graphs only) and add them to another figure. I would also strongly consider adding extended figure 3 or at least the subfigure 3c to the main text as the closing of the sheet is at the core of this work.

Our Answer: According to the referee remark, we completely reorganised the figures. Fig. 1 now illustrates the bio-inspired design of the device and the situation in which its deformations have been studied. Fig. 2 focuses on the equilibrium state of the capillary rise(s). Fig. 3 shows the capillary rises dynamics. Fig. 4 illustrates the situation in which the device is fully immersed in a liquid. And Fig. 5 illustrates the potential application of the device in the biomedical field and its ability to capture aqueous liquids and to achieve parallel tests. As required by the referee, we added the panel 3c of the Supplementary Figure 3 to Fig. 2. Moreover, information on the evolution of the flow speed across the deforming grooves, and of the hydraulic resistance –that were initially discussed only in the “methods” section– now appear in Fig. 3. Regarding the old Fig. 4, which corresponds to Fig. 5 now, we reduced the size of the old panel a, move panel c to Fig. 3, and we added a new panel (Fig. 5a) proving the ability to use our device to capture blood.

Reviewer 2 states: Finally, as mentioned, I have some comments about biomedical applications. Most importantly, it would be needed to specify for which kind of point-of-care applications it could be used. For blood analysis with immunoassays, reactions are carried out on the plasma only so that the centrifugation applied in the case of lab-on-a-disk devices for examples is needed and not to be suppressed. Those disks are very cheap and offer the possibility to have sequential reactions with stored reactants, valves, mixing and extraction processes... If compared to small glass/plastic capillaries used to sample blood as illustrated in Figure 4a, I am not sure that the storage argument is so strong (page 14) and the flexibility of the sheet could induce difficulties in manipulation compared to a rigid capillary. It is probably more appropriate to compare the proposed system to paper microfluidic. Here again, I am a bit missing arguments for the moment – especially if as I have no hydrophilic and environment friendly elastomers in mind. Could the authors better explain the advantages of their systems over established ones or lower their claims?

Our Answer: We thank the reviewer for the remark. Indeed, at this stage, we are not able to compare our device directly and quantitatively to existing approaches and thus prove the potential advantages of the approach. Such a quantitative comparison will take time and is the focus of a current effort in the team, following the patent application.

Changes to the manuscript. We completely reorganised the conclusion, lowered the claims and better explained what we believe are the two features of our device that could be useful for point-of-care applications: (i) a passive change in channel shape that allows for fast initial liquid transport and a protected liquid sample in the deformed state, (ii) an augmented volume captured thanks to deformability.

To go one step further, we added an experimental evidence that our device can be effectively used to capture blood (Figure 5a). This is enabled by carrying out a surface treatment of the initially hydrophobic silicone structure: first a plasma treatment is achieved, making the surface hydrophilic on short times, then the grooves are filled with a solution of serum albumin at 2wt%, and last, the serum albumin solution is dried by putting the imbibed structure in an oven at 50°C during one hour. After treatment the surface remains hydrophilic for long (at least one day) and can be used for the capture of any aqueous liquid. Making the device hydrophilic opens the perspective of depositing a solution of antibodies into the structure's grooves. After drying out the solution, the prepared structure could be used for blood testing (see Methods, section 'Surface treatment of the elastomer').

We also completed the quantitative comparison between our flexible device and rigid counterparts, which was done so far only in Figure 4 (Fig. 3 in the previous version of the manuscript).

- We added a Figure panel (Fig. 2e), in which we quantitatively compare the amount of liquid captured depending on the deformability of the device.
- We added a Figure panel (Fig. 3d), in which we show that, for the specific groove aspect ratio considered in this study, the hydraulic resistance increases by more than a factor 6 between the open and closed (deformed) state, allowing fast liquid transport in the early stages of fluid capture and a protected liquid sample after deformation.

Response to reviewer 3.

Reviewer 3 states: This proposed manuscript discusses a method for utilizing capillary fluid capture in an elastically deformable set of channels that is a biomimicry of the grooved tongue of a hummingbird. The authors compare and extend surrogate experiments to their theoretical models to provide insights into the dominant physics of the system.

According to the website,

Nature Communications is an open access, multidisciplinary journal dedicated to publishing high-quality research in all areas of the biological, health, physical, chemical, Earth, social, mathematical, applied, and engineering sciences. Papers published by the journal aim to represent important advances of significance to specialists within each field.

Overall, I think this proposed manuscript is within the scope of the Nature Communications. However, I do think that the manuscript would benefit from having some minor assistance with writing. I have provided some comments below to consider.

Our Answer: We thank Reviewer 3 for their positive feedback on the content of the article. We made a significant effort to reorganize the structure of the paper to improve its readability. Please find below our point-by-point answer to the comments of the reviewer.

Queries

Reviewer 3 states: I would imagine that an additional situation would exist where the wall length h is too short so that the channel system never forms closed triangles before the system forms a closed tube. Was this scenario explicitly discarded? Can the scenario be quickly addressed?

I would also imagine that an additional situation would exist where the wall length h is too large

so that the system never forms a closed tube [see Eqn. (8) such that $h > R$]. Was this scenario explicitly discarded? I believe that it is implied on Line [435]. Can the scenario be quickly addressed?

Our Answer: In the article, we chose to focus on the situation where the width of the system (or equivalently the number of grooves) is adapted in order to perfectly close into a tube. Hence, when the height of the ribs h and the width of the grooves d is changed, we adjust the number of grooves $n = 2\pi h/d$ (i.e. the width of the system $n(d + w)$) to ensure that it closes in a tube.

As the reviewer points out, if the number of grooves is smaller than n , the ribbon cannot close into a tube and rolls (if sufficiently soft) into a portion of a tube with each groove closed. The theory for groove closure developed in the article still holds in this case, the only difference is related to the second capillary rise, that does not occur.

If the number of grooves is larger than n , the ribbon rolls into itself and self-contact appears at the ribbon scale before the closure of the groove, leading to a more intricate and less regular shape.

Changes to the manuscript. We added a section ‘Varying the number of grooves’ in the methods to discuss these two scenarios together with experimental evidence and a Supplementary Figure (shown below).

Reviewer 3 states: Throughout the manuscript, the contact angle θ is assumed to be $\theta = \pi/2$. Yet, the authors seem to introduce a second contact angle $\theta_Y = 0$ without discussion or justification.

Our Answer: As we consider a perfectly wetting liquid the contact angle is indeed $\theta_Y = 0$ at the interface between the solid and the liquid. However, the wedge of the ribs is a singular point where the angle of the liquid-solid interface can take any value between 0 and $\pi/2$ if the liquid perfectly wets the surface. For the sake of simplicity, we assumed that the angle there is constant and reads $\theta = \pi/2$. Note that this simplifying assumption is not entirely rigorous, since the curvature of the interface must vary along the vertical axis to obey Laplace law. The good agreement between the model and the experiments validate the assumption.

Changes to the manuscript. We added a short discussion in the Methods, section ‘Geometry of a groove and of the air-liquid interfaces’, to better explain this point : “Note that this angle is compatible with a zero wetting contact angle (i.e., a perfectly wetting liquid). In fact, the wedge of the ribs is a singular point where the angle of the liquid-solid interface can range from 0 to $\pi/2$ if the liquid perfectly wets the surface. Here, an angle of $\pi/2$ corresponds to a zero contact angle relative to the top of the walls.” (lines 487-490).

Figure 2: **Variation of the Number of grooves.** In the first row, the number of grooves, n , is insufficient to close the sheet into a tube, resulting in the formation of a circular arc. The second row corresponds to the case studied in this article, where the sheet deforms into a tube with an outer radius of $R_{\text{out}} = h(1 + w/d)$. In the third row, an increase in the number of grooves, n , causes the structure to deform into a tube with a larger radius than $h(1 + w/d)$, where the top edges of all the ribs do not come into contact. Finally, the fourth row depicts a situation where the structure deforms into a spiral due to overlap.

Reviewer 3 states: How sensitive is the system to deviations from \$\mathbf{g} = -g\mathbf{e}_z\$?

Our Answer: The equation describing the flows in the grooves can easily be modified when the structure is inclined by an angle of Ψ with the gravitational axis by substituting ρg to $\rho g \cos(\Psi)$ in Eqs. (12) and followings. Thus, for small Ψ angle the equation remain unchanged up to order

2 in Ψ . It is similar for the torque balance equation derived in the article that also remains valid up to order 2 in Ψ . Hence, for small variation of the orientation of the device, we do not expect significant changes in the curling dynamics.

However, for larger values of Ψ , we expect that a variation of the sheet orientation impacts the curling dynamics of the sheet. Moreover, in that case the modification of the torque balance equation to take into account gravitational forces is far from being trivial and would require a proper study which is out of the scope of this article.

Major Comments

- The figure references are very, very confusing. Most figures do not appear to be introduced, e.g., Extended Data Figure 3, before they are shown. Line [62] introduced Fig. 3, but Fig. 2 was not introduced to that point.

Our Answer: We thank the reviewer for pointing out this issue. We completely reorganized the figures and made sure that each Extended Data Figure is properly referenced in the manuscript.

- Line [81] introduces ‘case (i),’ while Line [94] introduces ‘case (iii).’ Where is case (ii)?

Our Answer: We thank the reviewer for pointing out the issue. Case (ii) is now introduced alongside the case (iii) (line 99).

- Line [144] I would like to see the authors expand the introduction of the statement that “experience a symmetry breaking event [that is] commonly reported in elastocapillary phenomena.”

Our Answer: In elastocapillary systems where there is an array of slender structures, each structure is symmetrically pulled by capillary forces on each side. The deflection and coalescence of the structures, that eventually bend left or right results from a symmetry breaking instability. Such results have been largely discussed in a rich literature (eg. Ref. ⁶⁻⁸, see below). In our case, the finite bending stiffness of the walls between the grooves can explain the variability in groove closure observed in ribbons close to the complete closure transition.

Changes to the manuscript. We rephrased this sentence to better explain what we mean : “Note that near the closing transition ($1 \leq \Lambda \leq 8$), there is significant variability in the final capillary rise, ℓ_I^∞ . This variability, which is also visible in the second image of Fig. 2a, occurs within the same sheet (i.e., for the same Λ) and is not accounted for by our model, which assumes identical behavior for each groove. We attribute this variability to small differences in groove sizes, which can substantially affect the equilibrium height. Additionally, these size differences can result in uneven forces acting on either side of the wall, potentially causing bending and symmetry-breaking events. Such effects, commonly observed

in elastocapillary phenomena, greatly amplify variations in the groove cross-sections and, consequently, the equilibrium capillary rise height.” (lines 158-166).

- Between Lines [155] and [156], I would suggest introducing some section transition.

Our Answer: We thank the reviewer for the suggestion. Sections have been added to the manuscript.

- Line [192-193]: This statement is confusing, at best.

Our Answer: We reformulated the sentence and hope it is now clearer: “No limitation exists on the whole sheet width, which can be much larger than the capillary length while still deformed by capillary forces. This is because the capillary actuation is local and occurs at the scale of the grooves, satisfying $w < \ell_c$.” (lines (250-253))

- I did not vet the equations, which should be done. For instance, Eqn. (21): I am assuming that the prime indicates the spatial derivative. If so, I would write it explicitly, or I would explicitly define it, just as the dot is defined in Line [451].

Our Answer: Indeed the “prime” was indicating a spatial derivative. We now give the definition of the “prime” in Eq. (23) to make it clear.

Minor Comments

In general, grammar and punctuation could be improved in multiple locations. I have provided a few changes below; however, a number of unlisted instances also exist. Having a colleague provide feedback on the grammar and punctuation could be beneficial.

- Lines [4, 21, 494, 505]: Writing a compound verb requires the type of verb to be consistent. We corrected the sentences.
- Lines [52] and Fig 4 Caption: The inconsistent usage (and probable overuse) of dashes is distracting. We corrected the sentences.
- Fig. 1: I would suggest printing the manuscript on a black and white printer to see how distinguishable the plots will be. Seeing ‘light blue’ versus ‘dark. We changed the colors in the graphs to increase the contrast between the curves.
- Lines [67, 72, 156, 375, 427, 482, 506, 574] and Fig 4 Caption: Writing a sentence with two independent clauses that are separated by a coordinating conjunction required a comma to aid understanding. We corrected the sentences.

- Line [75]: Consider using ‘scenarios’ over ‘scenarii.’
We corrected the typo.
- Line between [127] and [128] and Lines [192, 369, 423, 478, 486, 513]: Using words such as ‘there’ and ‘it’ as subjects makes for confusing sentences. ‘Where’ is ‘there’? Further, using ‘it’ without an antecedent is not grammatically correct.
We corrected the sentences.
- Line [149, 215, 228, 493, 519+]: Using ‘[t]his’ or ‘those’ without an object is grammatically incorrect.
We did our best to correct the grammar.
- Line [189]: I do not think that they meant to use ‘elastica.’
The wording ‘elastica’ is used in the elasticity community to describe the nonlinear 2D theory of rods developed by Euler and Bernoulli. We replaced this jargon by “linear elastic theory”.
- Line [209]: I would suggest not using possessive language for a texture (e.g., “the textures’ geometries” should be “the geometries of the texture”).
We corrected the sentence.
- The bibliography needs vetted.
We corrected the errors we spotted in the bibliography.
- Line [399]: “... its area coincide[s] with ...”
Line [457]: “The capillary pressure i[s] then given ...”
Line [507]: “... branch of f [is] located ... ”
Line [514]: “... and β jump[s] from ...”
We thank the referee for spotting the typos. We corrected them.

References

1. Wells, R. E., Merrill, E. W. et al. Influence of flow properties of blood upon viscosity-hematocrit relationships. The Journal of clinical investigation **41**, 1591–1598 (1962).
2. Rand, P. W., Lacombe, E., Hunt, H. E. & Austin, W. H. Viscosity of normal human blood under normothermic and hypothermic conditions. Journal of applied physiology **19**, 117–122 (1964).
3. Yang, D., Krasowska, M., Priest, C., Popescu, M. N. & Ralston, J. Dynamics of capillary-driven flow in open microchannels. The Journal of Physical Chemistry C **115**, 18761–18769 (2011).

4. Kolliopoulos, P. et al. Capillary-flow dynamics in open rectangular microchannels. Journal of Fluid Mechanics **911**, A32 (2021).
5. Kim, J., Jung, Y. & Kim, H.-Y. Evaporative capillary rise. Physical Review Fluids **7**, L032001 (2022).
6. Bradley, A. T., Hewitt, I. J. & Vella, D. Bendocapillary instability of liquid in a flexible-walled channel. Journal of Fluid Mechanics **955**, A26 (2023).
7. Liu, X. et al. Capillary-force-driven self-assembly of 4d-printed microstructures. Advanced Materials **33**, 2100332 (2021).
8. Bico, J., Reyssat, É. & Roman, B. Elastocapillarity: when surface tension deforms elastic solids. Annual Review of Fluid Mechanics **50**, 629–659 (2018).

Point-by-point Response to the Reviewers

We thank the reviewers for their time and their feedback on our manuscript. Please find below our point-by-point responses to the reviewers' comments.

Response to Reviewer 1:

Reviewer 1 states: "No comments"

Response to Reviewer 2:

Reviewer 2 states: "I had two major comments related to (i) the organization and readability of the article and (ii) the strength of the claims regarding potential applications for Point of Care (POC). Regarding the first point (i), the authors have carefully addressed the questions and reorganized the manuscript and figures, which has significantly improved the paper. I still have a comment about figure readability — ℓ_I (the letter, not the symbol when plotted in a graph) is barely visible on my screen. Perhaps using bold fonts or a slightly darker blue would improve visibility. Furthermore, the authors have significantly toned down their claims, making them more aligned with the current state of their research. However, presenting the need to make a hydrophobic elastomer hydrophilic through a subsequent surface treatment — whose effectiveness is limited to just one day — as a possible solution rather than a potential challenge remains misleading. In real applications, where production costs, shelf life, and reproducibility are crucial, this remains a critical issue and I would appreciate if a remark could be added to underline this limitation.

I have no further comments beyond these two points, which could quickly be addressed prior to publication."

Our Answer: We thank the reviewer for their thorough review of our paper. According to the reviewer remark we changed the colour of ℓ_I symbols in the figures to increase the readability. In accordance with the reviewer's suggestions, we have added a concluding sentence to the article discussing the potential need to modify the device material for application in blood diagnostics.

Response to Reviewer 3:

Reviewer 3 states: "I appreciate the responses to my inquiries."

Our Answer: We thank the reviewer for their comment.

Review: Elastocapillary sequential fluid capture in hummingbird-inspired grooved sheets

This proposed manuscript discusses a method for utilizing capillary fluid capture in an elastically deformable set of channels that is a biomimicry of the grooved tongue of a hummingbird. The authors compare and extend surrogate experiments to their theoretical models to provide insights into the dominant physics of the system.

According to the website,

Nature Communications is an open access, multidisciplinary journal dedicated to publishing high-quality research in all areas of the biological, health, physical, chemical, Earth, social, mathematical, applied, and engineering sciences. Papers published by the journal aim to represent important advances of significance to specialists within each field.

Overall, I think this proposed manuscript is within the scope of the *Nature Communications*. However, I do think that the manuscript would benefit from having some minor assistance with writing. I have provided some comments below to consider.

Queries

- I would imagine that an additional situation would exist where the wall length h is too short so that the channel system never forms closed triangles before the system forms a closed tube. Was this scenario explicitly discarded? Can the scenario be quickly addressed?
- I would also imagine that an additional situation would exist where the wall length h is too large so that the system never forms a closed tube [see Eqn. (8) such that $h > R$]. Was this scenario explicitly discarded? I believe that it is implied on Line [435]. Can the scenario be quickly addressed?
- Throughout the manuscript, the contact angle θ is assumed to be $\theta = \pi/2$. Yet, the authors seem to introduce a second contact angle $\theta_Y = 0$ without discussion or justification.
- How sensitive is the system to deviations from $\underline{g} = -g\underline{e}_z$?

Major Comments

- The figure references are very, very confusing. Most figures do not appear to be introduced, e.g., Extended Data Figure 3, before they are shown. Line [62] introduced Fig. 3, but Fig. 2 was not introduced to that point.
- Line [81] introduces ‘case (i),’ while Line [94] introduces ‘case (iii).’ Where is case (ii)?
- Line [144] I would like to see the authors expand the introduction of the statement that “experience a symmetry breaking event [that is] commonly reported in elastocapillary phenomena.”
- Between Lines [155] and [156], I would suggest introducing some section transition.
- Line [192-193]: This statement is confusing, at best.
- I did not vet the equations, which should be done. For instance, Eqn. (21): I am assuming that the prime indicates the spatial derivative. If so, I would write it explicitly, or I would explicitly define it, just as the dot is defined in Line [451].

Minor Comments

In general, grammar and punctuation could be improved in multiple locations. I have provided a few changes below; however, a number of unlisted instances also exist. Having a colleague provide feedback on the grammar and punctuation could be beneficial.

- Lines [4, 21, 494, 505]: Writing a compound verb requires the type of verb to be consistent.
- Lines [52] and Fig 4 Caption: The inconsistent usage (and probable overuse) of dashes is distracting.
- Fig. 1: I would suggest printing the manuscript on a black and white printer to see how distinguishable the plots will be. Seeing ‘light blue’ versus ‘dark blue’ is difficult.
- Lines [67, 72, 156, 375, 427, 482, 506, 574] and Fig 4 Caption: Writing a sentence with two independent clauses that are separated by a coordinating conjunction required a comma to aid understanding.
- Line [75]: Consider using ‘scenarios’ over ‘scenarii.’
- Line between [127] and [128] and Lines [192, 369, 423, 478, 486, 513]: Using words such as ‘there’ and ‘it’ as subjects makes for confusing sentences. ‘Where’ is ‘there’? Further, using ‘it’ without an antecedent is not grammatically correct.
- Line [149, 215, 228, 493, 519+]: Using ‘[t]his’ or ‘those’ without an object is grammatically incorrect.
- Line [189]: I do not think that they meant to use ‘elastica.’
- Line [209]: I would suggest not using possessive language for a texture (e.g., “the textures’ geometries” should be “the geometries of the texture”).
- The bibliography needs vetted.
- Line [399]: “... its area coincide[s] with ...”
- Line [457]: “The capillary pressure $i[s]$ then given ...”
- Line [507]: “... branch of f [is] located ... ”
- Line [514]: “... and β jump[s] from ...”